palaeontology

integumentary evolution, Recumbirostra, Carboniferous, Mazon Creek, scale ultrastructure, Amniota

**Author for correspondence:**
Arjan Mann
e-mail: arjan.mann@carleton.ca

# *Joermungandr bolti*, an exceptionally preserved 'microsaur' from the Mazon Creek Lagerstätte reveals patterns of integumentary evolution in Recumbirostra

Arjan Mann, Ami S. Calthorpe and Hillary C. Maddin

Department of Earth Sciences, Carleton University, 2115 Herzberg Laboratories, 1125 Colonel By Drive, Ottawa, Ontario, Canada K1S 5B6

AM, 0000-0002-3511-0635; HCM, 0000-0002-6969-4907

The Carboniferous Pennsylvanian-aged (309–307 Ma) Mazon Creek Lagerstätte produces some of the earliest fossils of major Palaeozoic tetrapod lineages. Recently, several new tetrapod specimens collected from Mazon Creek have come to light, including the earliest fossorially adapted recumbirostrans. Here, we describe a new long-bodied recumbirostran, *Joermungandr bolti* gen. et sp. nov., known from a single part and counterpart concretion bearing a virtually complete skeleton. Uniquely, *Joermungandr* preserves a full suite of dorsal, flank and ventral dermal scales, together with a series of thinned and reduced gastralia. Investigation of these scales using scanning electron microscopy reveals ultrastructural ridge and pit morphologies, revealing complexities comparable to the scale ultrastructure of extant snakes and fossorial reptiles, which have scales modified for body-based propulsion and shedding substrate. Our new taxon also represents an important early record of an elongate recumbirostran bauplan, wherein several features linked to fossoriality, including a characteristic recumbent snout, are present. We used parsimony phylogenetic methods to conduct phylogenetic analysis using the most recent recumbirostran-focused matrix. The analysis recovers *Joermungandr* within Recumbirostra with likely affinities to the sister clades Molgophidae and Brachystelechidae. Finally, we review integumentary patterns in Recumbirostra, noting reductions and losses of gastralia and osteoderms associated with body elongation and, thus, probably also associated with increased fossoriality.

# 1. Introduction

Recent resurgence in the study of certain 'microsaurs' known as recumbirostrans has resulted in their renewed relevance in the origin of amniotes [1,2]. Debates specifically concern whether Recumbirostra belong to the amniote stem group [3,4], or are instead crown amniotes derived from a reptilian ancestor [2]. The timing of the origin of amniotes is earmarked at the early Pennsylvanian, with the oldest unambiguous amniotes residing in the Bashkirian-aged (approx. 318 Ma) strata of Joggins, Nova Scotia [5,6]. Already at this stage, and onward through the Permo-Carboniferous, recumbirostrans are morphologically diverse, showing a range in degree of development of a variety of stereotypical cranial and postcranial adaptations for a fossorial lifestyle [7–13]. Thus, the inclusion of Recumbirostra into Eureptilia would redefine the nature of the radiation of the groups and reveal details of the ecology and diversity in some of the earliest phases of reptile evolution. Recent studies [13–16] on tetrapod fossils from the Moscovian-aged (309–307 Ma) Mazon Creek Lagerstätte have revealed a diverse 'recumbirostran-amniote' assemblage. Because Mazon Creek often preserves fossils of entire organisms, including soft-tissue structures entombed within siderite concretions, a diverse array of baupläne have been preserved, questioning past ideas about terrestrial tetrapod diversity in Carboniferous ecosystems ([17–21]; but see [22]).

Here, we describe *Joermungandr bolti* gen. et sp. nov., a new long-bodied recumbirostran from Mazon Creek, Francis Creek Shale, Illinois (figures 1–3). *Joermungandr* provides an ecomorphological intermediate between the extreme body elongation and limb reduction seen in molgophid recumbirostrans such as the contemporaneous *Infernovenator* and the short-bodied, robustly built brachystelechid recumbirostran *Diabloroter* [13,14]. *Joermungandr* is exceptionally preserved, at the higher end of the quality scale for Mazon Creek fossils. It includes alongside the full skeleton visible in dorsal and ventral aspects, a detailed soft tissue impression of the body complete with scales. Detailed examination of the scales of *Joermungandr* reveals a unique ultrastructural pattern, and provides the first comprehensive description of scale morphology of this kind in a Palaeozoic tetrapod (figure 4).

# 2. Material and methods

Specimens were studied at: Augustana College's Fryxell Geology Museum (ACFGM), Rock Island, USA; American Museum of Natural History (AMNH), New York, USA; Carnegie Museum of Natural History (CM), Pittsburgh, USA; Denver Museum of Nature and Science (DMNH), Denver, USA; Field Museum of Natural History (FMNH), Chicago, USA; University of Kansas Natural History Museum (KUVP), Lawrence, USA; University of Nebraska State Museum (UNSM), Lincoln, USA; Smithsonian Institution (USNM), Washington, DC, USA; and Yale Peabody Museum (YPM), New Haven, USA. Additional comparative specimens from the British Museum of Natural History (BMNH), London, UK, the Harvard Museum of Comparative Zoology (MCZ), Cambridge, USA, Museum für Naturkunde (MB), Berlin, Germany, were compared based on casts, latex peels and existing literature.

Only one latex peel was made of the dorsal aspect of the skeleton, and is catalogued with the original material at the FMNH. This latex cast was used to describe the dorsal cranial and postcranial elements; however, the ventral half of the skeleton is described solely off the corresponding counterpart. Photography was conducted using a Sony Alpha ILCE 5000 camera, F3.5 lens. All figures were drawn and formatted in Photoshop CS6 (Adobe, San Jose, CA, USA). For phylogenetic methodology, see the Phylogenetic analysis section.

In order to interpret features of the morphology of the scales preserved on *J. bolti*, extant comparative data were examined, consisting of squamate scales belonging to a number of terrestrial, arboreal and fossorial forms housed in the research collections of the Canadian Museum of Nature, Gatineau, Quebec (CMN). A list of selected extant squamate specimens can be found in the electronic supplementary material. Scanning electron microscopy (SEM) was conducted at the Canadian Museum of Nature research facility using a JEOL 6610LV SEM.

## 2.1. Anatomical abbreviations

ang, angular; bo, basioccipital; co, co-ossified occipital and otic elements (*os basale*); d, dentary; ec, ectopterygoid; f, frontal; fe, femur; fib, fibula; gb, gastric bolus; gs, gastralia; h, humerus; il, ilium; j, jugal; l, lacrimal; m, maxilla; mt, metatarsal; n, nasal; p, parietal; pas, parasphenoid; pf, pineal foramen; pmx, premaxilla; pob, postorbital; pof, postfrontal; prf, prefrontal; pt, pterygoid; qj, quadratojugal; so, supraoccipital; sp, splenial; sq, squamosal; sv, sacral vertebra; t, tabular; tib, tibia; u, ulna; v, vertebrae.

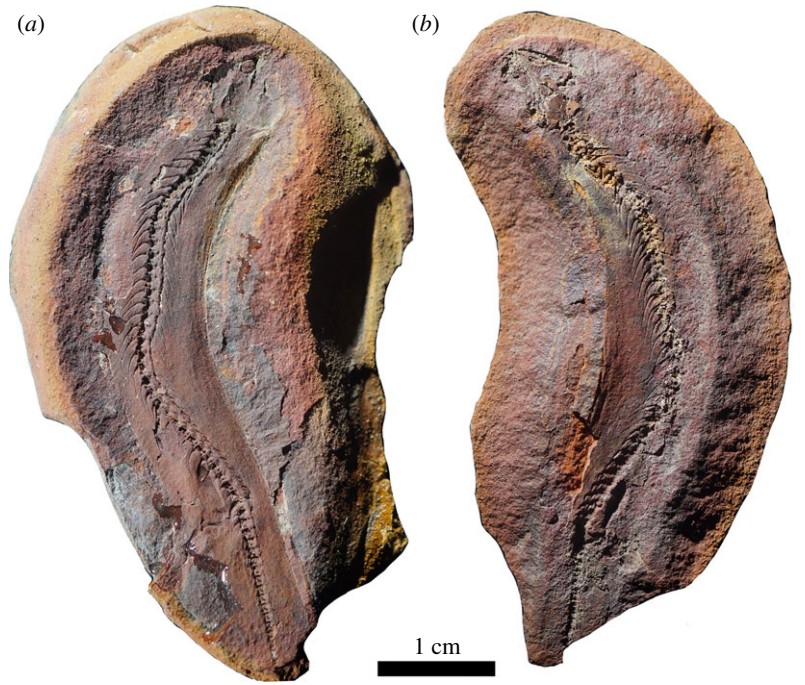

**Figure 1.** Photographs of the holotype of *J. bolti* gen. et sp. nov. (FMNH 1309). (*a*) The part specimen showing the dorsal view; (*b*) the counterpart specimen showing the ventral view.

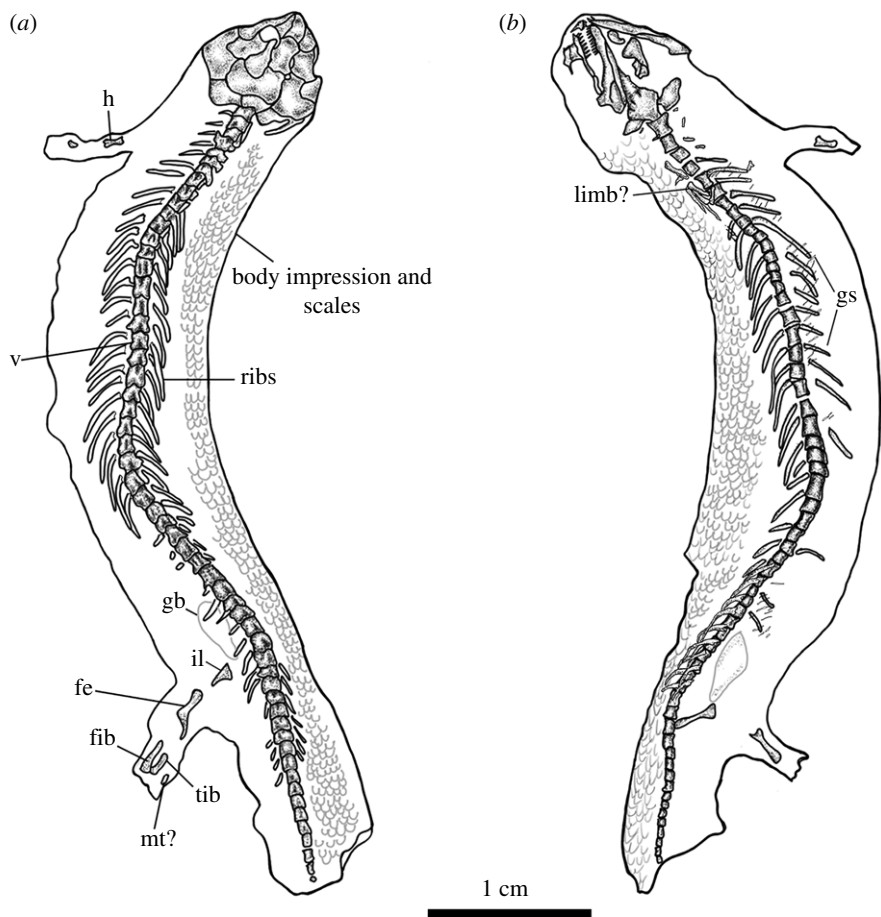

**Figure 2.** Illustrations of the holotype of *J. bolti* gen. et sp. nov. (FMNH 1309) (*a*), dorsal and (*b*), ventral aspects of the holotype of *J. bolti* gen. et sp. nov. (FMNH 1309). fe, femur; fib, fibula; gb, gastric bolus; gs, gastralia; h, humerus; il, ilium; mt, metatarsal; tib, tibia, vertebrae.

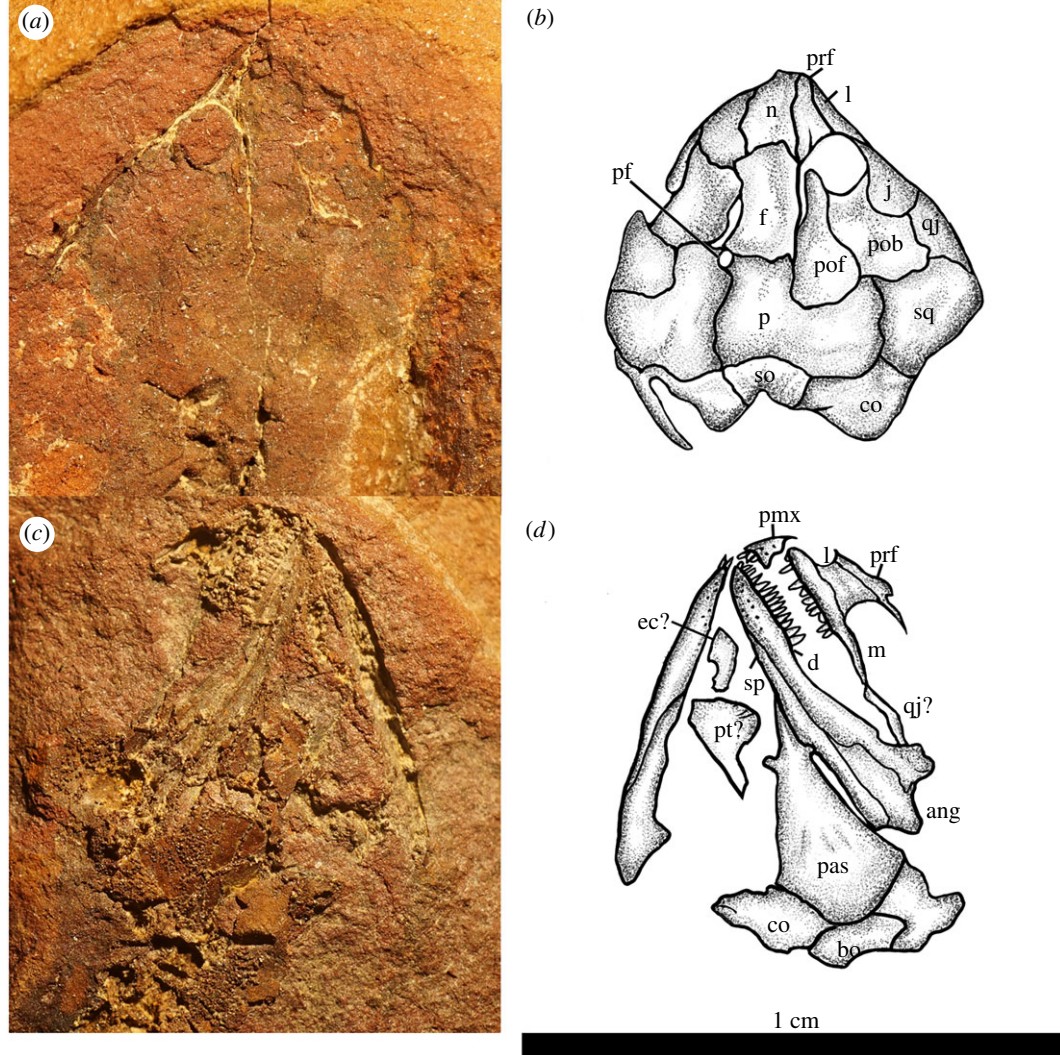

**Figure 3.** Illustrations of the (a,b) dorsal and (c,d) ventral aspects of the skull of *J. bolti* (FMNH 1309). Illustrations have been digitally mirrored in Photoshop to show true anatomical orientations. ang, angular; bo, basioccipital; co, co-ossified occipital and otic elements (os basale); d, dentary; ec, ectopterygoid; f, frontal; j, jugal; l, lacrimal; m, maxilla; n, nasal; p, parietal; pas, parasphenoid; pf, pineal foramen; pmx, premaxilla; pob, postorbital; pof, postfrontal; prf, prefrontal; pt, pterygoid; qj, quadratojugal; so, supraoccipital; sp, splenial; sq, squamosal.

## 2.2. Systematic palaeontology

Tetrapoda [23]
Recumbirostra [24]
*Joermungandr bolti* gen. et sp. nov.
   (Figures 1–4)
**Zoobank LSID (genus):** urn:lsid:zoobank.org:act:E6D33CC7-7DC0-4A83-9B68-79156D1D9B59.
**Zoobank LSID (species):** urn:lsid:zoobank.org:act:9B06DE36-A6F5-4821-9A07-81C2151AF0CD.
**Holotype.** FMNH 1309, part and counterpart of a siderite concretion containing a virtually complete skeleton and soft body impression in dorsal and ventral views.
**Locality and Horizon:** Mazon Creek, Grundy County, IL, USA. Francis Creek Shale, above the Morris (no. 2) Coal, Carbondale Formation, Middle Pennsylvanian (Moscovian).
**Etymology.** 'Joermungandr' the Swedish phoneme of 'Jörmungandr' (gender: masculine) the name of the serpent that dwells in the 'Midgard Sea' from Norse mythology. The specific epithet '*bolti*' is in honour of the late palaeontologist John R. Bolt.
**Diagnosis.** A small recumbirostran with the following unique combination of characters: 40 presacral vertebrae; dentary and maxillary tooth rows terminate anteriorly, occupying approximately half the length of each respective element; short snout; large rectangular parietal that reaches the postorbital

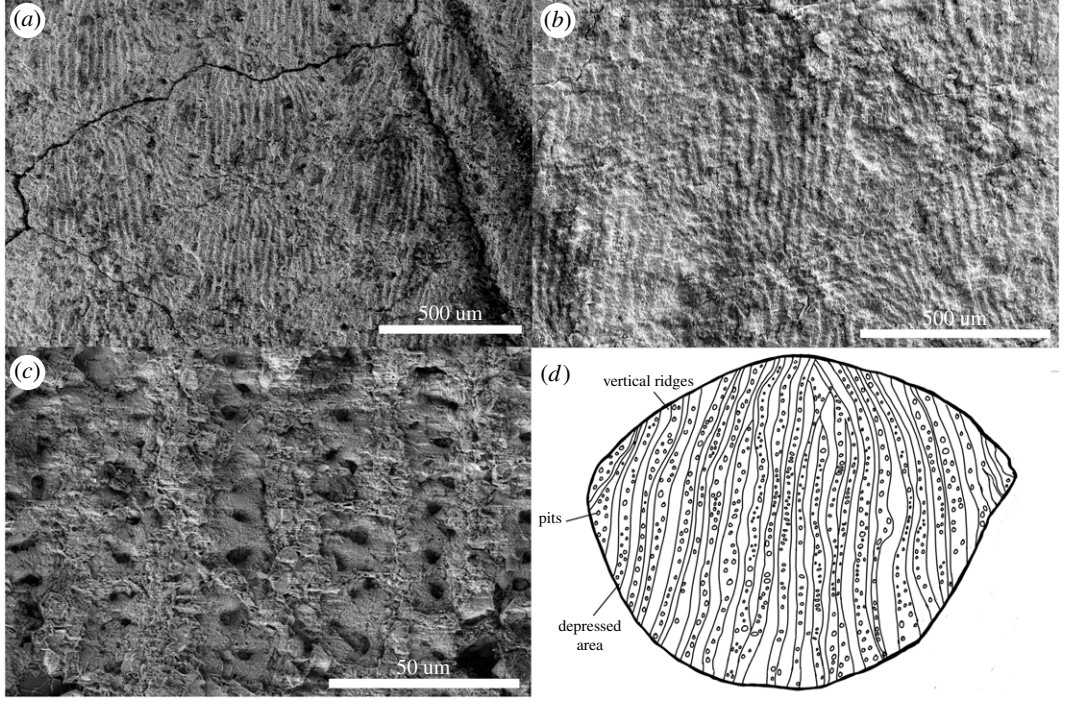

**Figure 4.** (a–c) Scanning electron micrographs (SEMs) of *J. bolti* (FMNH 1309), showing progressively closer structures of the dermal scales. (d) An idealized illustration of a single dermal scale (not to scale).

and squamosal; large quadrangular postfrontal that invades the anterior margin of the parietal; co-ossification of otic capsules and occiput; distinct supraoccipital; body covered in uniform scales; scales ornamented with linear ridges and unevenly distributed pits.

**Comments.** The 'microsaur' *Odonterpeton triangulare* [25] from Linton, Ohio, was placed by Carroll and Gaskill [26] into 'Microbrachomorpha'. However, the anatomy of this taxon has not been properly reassessed since, and thus, comparisons between it and recumbirostrans are limited. Our own personal examination of original latex peels of *Odonterpeton* (created by the late Donald Baird) reveals general similarities in vertebral architecture and cranial morphology to recumbirostrans. *Odonterpeton* is small and bears the most similarity to juvenile molgophids and brachystelechids. There also appears to be a number of similarities between *Joermungandr* and *Odonterpeton*, including general size, widened parietals, large triangular shape of the parasphenoid, anterior position of the orbits and short snout. These may indicate a close relationship between these two taxa; however, *Joermungandr* can at least be distinguished from *Odonterpeton* by a well-developed cheek region (which is probably emarginated in *Odonterpeton* in a similar manner to lysorophians), a large postfrontal and rounded skull shape. Additionally, the anterior limbs in *Odonterpeton* are diminutive but well ossified, whereas the anterior limbs of *Joermungandr* appear proportionally longer and less well ossified.

# 3. Description

## 3.1. Cranial anatomy

The dorsal cranium is well preserved; however, it is folded-over on the left side revealing the right-lateral surface including the orbital and snout morphology (figure 3). The left orbit is not visible because it is hidden under the dorsal skull on that side. The right orbit, which is present, shows a slightly raised orbital rim. There is no cranial ornamentation present on the dorsal skull elements, which may indicate the animal is a juvenile [27]. Like many recumbirostrans, the entire skull is robustly ossified, short and somewhat rounded as the skull transitions from the dorsal to lateral surfaces. The dorsal outline of the cranium is roughly triangular. *Joermungandr* bears the most similarity in gross skull morphology to round-headed ecomorphs such as *Carrolla* and *Quasicaecilia* [8,10,28].

The paired nasals are located at the anteriormost portion of the preserved dorsal skull and are rectangular in shape (figure 3). The right nasal is better preserved than the left, the latter of which is partially obstructed by the right nasal, but also appears to dip under the skull in this area. There is no

indication that the dorsal surfaces of the nasals were invaded by the dorsal processes of the premaxillae. The nasals are longer than they are wide. They are shorter than the frontals in length but still quite wide posteriorly and taper slightly anteriorly. The nasals appear to curve ventrally along with the rest of the snout (i.e. prefrontal, lacrimal).

The frontals are paired, rectangular and isolated from the orbital margin by long anterior processes of the postfrontals and a smaller contribution from the prefrontals (figure 3). The morphology of the interfrontal suture cannot be discerned because the right frontal is slightly overlapping the left. However, it is likely that the suture was straight based on medial margins of the frontals that are visible. The frontals only very slightly expand in width posteriorly towards the parietals. The frontals contact the parietals at a simple, straight suture.

The parietals are paired, massive elements that are great in width and slightly longer in length than the frontals (figure 3). They are roughly rectangular, invaded anterolaterally by the main body of the postfrontal. A small pineal foramen is present between the two parietals and is anteriorly located similar to that in *Diabloroter* [14]. While it does appear that the pineal foramen contacts the frontals, this is most likely a result of post-mortem shifting of the frontals to slightly overlap the parietals. Generally, the large parietals are reminiscent of brachystelechids.

The occipital margin is undulating in profile and mostly dorsally exposed. The occipital and otic elements are co-ossified into a single ossification (often called an *os basale* in brachystelechids, see [10]), with only a distinct rectangular supraoccipital element visible. The ventral margin of the supraoccipital is excavated for the dorsal margin of the foramen magnum. The large co-ossified mass connects to the squamosals anterolaterally. The squamosals are large and irregularly shaped elements. Each squamosal contacts the corresponding parietal medially at an undulating suture; anteriorly, there are also contacts with both the postorbital and quadratojugal creating a similarly undulating suture.

The postorbitals are large, quadrangular-shaped elements that are slightly expanded posteriorly (figure 3). Anteriorly, the postorbitals are excavated by the respective posterior orbital margins. Anterolaterally, the postorbitals are excavated by the jugals which invade the postorbitals at a rounded suture. A relatively expansive, sub-rectangular-shaped lateral exposure of the quadratojugal (equal in size to the jugal) is visible between the jugal and the squamosal on the right side of the cranium (figure 3). On the left side, the quadratojugal is probably represented by a thin strip of poorly preserved bone that narrowly contacts the maxilla anteriorly. Also on the right side of the cranium, a teardrop-shaped jugal is present, with a wide expansive ramus on the cheek that tapers to a small elongated anterior process. It is inferred that the anterior process of the jugal rests atop of the maxilla and smoothly attenuates atop the ventral process of the lacrimal forming most of the ventral border of the orbit.

The lacrimals are long and sub-rectangular in shape. They extend from the nares to the orbit where each has a small dorsal and ventral orbital process (figure 3). The lacrimals are visible in the dorsal profile on the right side of the skull and in lateral profile on the counterpart. Dorsal to the lacrimals are the prefrontals, represented on either side of the cranium. The right prefrontal is better represented, where it is seen to be a long, triangular-shaped element that leads into the nares at its narrowest point. There are prominent posterior processes on the prefrontals that meet the anterior processes of each postfrontal. Together, these exclude the corresponding frontal from the dorsal orbital margin.

The upper jaw elements preserved include the maxilla and the premaxilla that are only preserved on the left side of the skull (figure 3). The maxilla is long, extending to just beyond the posterior limit of the orbit (figure 3). There is a low facial process and a long posterior process. Interestingly, the teeth do not extend onto the posterior process but rather extend from the anteriormost end to just below the anterior orbital margin. There are approximately 11–12 tooth positions on the maxilla. The teeth that are in place are simple and conical in morphology. The premaxilla is only visible in lateral aspect revealing a slightly recurved dorsal process that is not flattened or as drastically recurved as in other recumbirostrans (figure 3). The dorsal process is also very short and probably did not extend onto the dorsal skull. The premaxilla appears to bear around six small tooth positions. The tooth morphology is the same as that of the maxilla.

The lower jaws are well represented on the counterpart showing the ventral aspect of the specimen (figure 3). The dentaries are represented by both the right dentary in ventral aspect and the left dentary which has shifted into lateral aspect. The anterior ramus of the dentary is long and slender, and there is a raised coronoid process posteriorly. This process is comparable in height to that of some gymnarthrids such as *Cardiocephalus* [26]. The dentary appears ornamented with small innervation pits. The angular is preserved on the left lower jaw, but is not preserved well enough to discern its anatomy. A single long splenial is observed buttressing the dentary, best represented on the left lower jaw as well. There are 14–

16 tooth positions preserved on the left dentary of *Joermungandr* (figure 3). Each tooth has a conical morphology with a simple point at the apex. Both the tooth rows of the dentaries and maxillae appear to be short in comparison with the length of each element, with preserved teeth occupying approximately only the anterior half of each element's length.

Preserved on the counterpart (ventral aspect), between the lower jaws, are elements of the palate and braincase (figure 3). Most of the palatal elements are not preserved well enough to confidently assign to a specific bone. There is a small, quadrangular element adjacent to the right dentary, but this element does not bear any distinguishing features. This element may either be the right palatine or ectopterygoid. A larger palatal element posterior to the unidentified element is probably part of the right pterygoid, and bears a process that is probably part of the quadrate branch of the pterygoid. Of the braincase, there is a large parasphenoid with small basipterygoid processes and a somewhat short but thin cultriform process. It is possible that the anterior portion of the cultriform process is slightly obscured by the left mandible. Posterior to the parasphenoid, there is a small quadrangular-shaped basioccipital preserved in the ventral aspect that is flanked by the co-ossified occipital elements on either side.

## 3.2. Postcranial anatomy

Nearly, the entire postcranial anatomy is preserved in the part and counterpart (figures 1 and 2); however, the distalmost area of the limbs are obscured by a dissolution gradient around the outer margins of the concretion. There are approximately 40 presacral vertebrae (figure 2). Whereas the atlas is preserved, no morphological features can be discerned, but the axis centrum is visible and is slightly expanded compared to the other cervical centra. Most of the trunk vertebrae are of a typical recumbirostran morphology, being represented by robust quadrangular centra. The preserved tail appears short compared to the length of the trunk, with approximately 17 caudal vertebrae that taper in size drastically to a very small terminal caudal vertebra. The dorsal ribs are well preserved and show a dicephalic rib head morphology. The ribs are thin and taper distally as in most recumbirostrans. Starting from the cervical region, the ribs gradually increase in length moving posteriorly down the axis before shortening in length again towards the sacral region. The caudal ribs are preserved up to the 10th caudal vertebra. They are straighter than those of the dorsal vertebrae in their morphology and also decrease in length posteriorly. On the counterpart, which contains the ventral aspect, there are a series of gastralia that appear reduced in both morphology and surface area occupied on the ventrum. Each gastralium is thin, rod-like and tapers on each end to articulate with the adjacent gastralium. Instead of a full matting of chevrons, together the gastralia are arranged into long threads stretching across the ribs. This reduced condition is also seen in *Batropetes* and *Diabloroter* [14]. Finally, it is important to clarify that the gastralia are preserved overlaying a series of ventral dermal scales, suggesting they are internalized structures, similar to gastral ribs of living amniotes ([29]; see discussion below).

The limb osteology is not particularly well preserved but shows a left forelimb with a poorly ossified humerus and an unidentified zeugopodial element (figure 2). Parts of the left and right hindlimb are preserved on either side. Overall, the hindlimb appears better ossified than the forelimb. The femur is well developed, long and rod-like in shape. The tibia and fibula are also well developed, but they are short and moderately bowed. On the left side, there is a singular tarsal element. There may have been more of the pes ossified in life, but this is obscured by the dissolution artefact around the edges of the concretion. There is no ossification of the pectoral region whatsoever, and there is very little of the pelvic girdle which is only represented by a sliver of partial ilium.

## 3.3. Soft tissue preservation and scales

*Joermungandr bolti* preserves a fair amount of rarely preserved features, including soft body impressions, a gastric bolus and numerous scales of which the tissue composition is unclear (figures 1 and 2). The integumentary impression of the body reveals a long, tubular body plan. There is a slight expansion in the width of the animal at the base of the skull, but otherwise, there is no constriction in the cervical region. Instead, the integumentary impression is consistent in width from the cervical to sacral region (figure 2). The postcranial skeleton occupies approximately one-third of the body width. Integumentary impressions around the limbs show somewhat long limbs, despite being incompletely ossified. The tapering of the tail vertebrae probably indicates the presence of a short tail. This tail appears to be expanded and rounded, similar to the morphology of the tails of some modern geckos and some skinks, which use their tails for fat storage [30,31] (figure 2).

*Joermungandr* also preserves a full suite of dorsal, flank and ventral scales. Scales are oval-shaped (wider than long) and arranged into linear rows that are overlapping, but not significantly so. Individual scales are small (approx. 500 µm in width) and nearly impossible to see with the unaided eye. The scales are the same size, shape and thickness on both dorsal and ventral parts of *Joermungandr*. Similarly, there is little observable difference between anterior and posterior body scales. The scales possess a unique iteration of 'microsaurian' ornamentation (figure 2). Under a light microscope, the scales appear to have a clam shell-like (or scallop-like) pattern on their surfaces caused by a series of longitudinal ridges that fan out slightly towards the lateral edges of each scale (presumably similar to the 'net-like' pattern noted by Carroll & Gaskill [26]). There are no thickened or fimbriated regions on any of the scales, unlike those of some other 'microsaurs' [26]; A.M. 2019, personal observation). SEM images of the dorsal aspect of the cranium reveal the presence of scales in this region. The cranial scales of *Joermungandr* differ slightly from the trunk scales in that they are sparsely pitted and lack the raised ridges that form the patterns present on the body and appendicular scales (see electronic supplementary material, for SEM image of cranial scales). This indicates that the cranial scales were smoother than the body scales, which is also the case in the head-shield scales of living fossorial reptiles [32].

Using SEM, we were able to observe ultrastructural details of the scales preserved in *Joermungandr* (figure 4). The SEM revealed a rugose texture of the longitudinal ridges and additional features in the trenches between (figure 4). Numerous tiny pits are seen within the floor of each trench. In many cases, there is a single row of pits, but in some places, the pits are more unevenly distributed, resulting in regions with partial double rows of pits between ridges. Alternatively, if these are dermal scales composed of bone, it is possible that the trenches are actually raised ridges on the scales, and that the pits are denticle-like protrusions—however, no scale morphology was discernible from the latex peels to confirm this configuration.

Directly relevant for comparison are the full suite of scales found associated with a specimen referred to as *Hyloplesion* (FMNH PR 981)—another exceptionally preserved 'microsaur' from Mazon Creek. The scales of FMNH PR 981 differ from those of *Joermungandr* in a few aspects, including their profile which is quadrangular and the orientation of the ridged ornamentation that is not aligned with the anteroposterior axis of the animal (see fig. 130 in Carroll & Gaskill [26]). Differences in scale morphology, including ultrastructural anatomy, probably reflect taxonomic differences among 'microsaurs' and may be useful for species diagnoses.

# 4. Phylogenetic analysis

We explored the phylogenetic relationships of *J. bolti* using a modified version of the recent matrix of Pardo *et al.* [2], which provides the most up-to-date matrix for assessing recumbirostran interrelationships. The matrix used has been taxonomically modified in recent studies by Mann & Maddin [13], Mann *et al.* [28] and Gee *et al.* [33]. These modifications were retained in the current analysis. We performed a parsimony analysis using PAUP software v. 4.0b10 [34] and *Eusthenopteron* was specified as the outgroup. We used the heuristic search option with the TBR search algorithm and 1000 random addition sequence replicates. Maxtrees was set at 10 000, and automatically increased by 100. All characters were equally weighted. All multistate scores were treated as polymorphic. All ambiguous character states were resolved using the ACCTRAN setting. Indices of goodness of fit of the character data to the topology (e.g. consistency index (CI), retention index (RI), rescaled consistency index (RC) and homoplasy index (HI)) were calculated in PAUP. To assess support of internal nodes, bootstrap values were calculated using the fast stepwise addition option with 1000 replicates.

The parsimony analysis recovered 270 most parsimonious trees (MPT), each with 1850 steps (CI = 0.306; HI = 0.748; RI = 0.649; RC = 0.199) (see electronic supplementary material for all phylogenetic analyses; S1–S3). The majority rule consensus of the results recovered *Joermungandr* as sister taxon to a clade that consists of a sister clade relationship between molgophids and brachystelechids. The majority rule consensus also recovers most previously reported relationships (e.g. [2]). The strict consensus of the results recovered *Joermungandr* in a polytomy within Recumbirostra. This polytomy includes: *Cardiocephalus peabodyi*; *Cardiocephalus sternbergi*; *Pariotichus brachyops*; *Euryodus dalyae*; *Euryodus primus*; *Proxilodon bonneri*; *Huskerpeton englehorni*; *Rhynchonkos stovalli*; a clade containing *Llistrofus* to the exclusion of ostodolepids; a clade where molgophids are sister taxa to brachystelechids; and finally, a clade containing the sister taxa *Dvellecanus carrolli* and *Aletrimyti*

*gaskillae*. This relationship is supported by five characters: 17(0), 28(2), 47(2), 48(0), 53(1), 115(0), 139(1) and 267(0) (all character numbers refer to those used in the analysis of [2]). The bootstrap tree and associated values can also be found in the electronic supplementary material (S3).

# 5. Discussion

## 5.1. Homology and terminology of integumentary structures in Recumbirostra

Despite the exceptional preservation of scales and other integumentary structures in a variety of Palaeozoic tetrapod groups (e.g. temnospondyls, 'microsaurs', reptiles), rather few studies on the diversity and evolutionary importance of these structures exist (see [35–37]). Witzmann [35] and Hook [38] provided useful discussions of the terminology applied to early tetrapod scales and scale-like ossifications in the palaeontological literature. These authors settled on a preferred terminology based on structural differences that is followed here: 'osteoderms', plate-like dermal ossifications that can bear pits on the outer surface; 'dermal scales', thinner than osteoderms but still composed of bone, ovular or round in profile, and can be overlapping; and 'gastralia', ventrally located elongate ossifications often collectively arranged into a chevron pattern. The thin, rod-like gastralia of early amniotes are probably derived from the dermal gastral scales of basal tetrapods [35,39–41]. Furthermore, these are probably deeply homologous with the gastralia (gastral ribs) in extant amniotes. In crocodilians and *Sphenodon*, the gastralia develop within the dermis and are secondarily integrated during ontogeny into the abdominal musculature ([29,40,42]; but also see [43]). The scales (dorsal, flank and ventral) of many recumbirostran 'microsaurs' predominantly fall into the dermal scale classification, as they appear to be composed of bone, are thin and arranged in overlapping clusters. However, the scales found on *Joermungandr* appear aberrant in their detailed ultrastructural morphology and in their preserved association with soft tissue impressions. As such, it is possible that these scales are keratinous. This would offer an explanation for the uncanny ultrastructural resemblances between these and other 'microsaur' scales (including Recumbirostra) and those of extant lepidosaurs [44–48]. Alternatively, the scales of *Joermungandr* may represent a combination of bone and keratinous scale covering. This could explain the roughened texture of the ridges (maybe the keratinous portion) residing atop a pitted base (maybe the bone portion). Characterizing the histology of these recumbirostran scales may provide insights into the homology of integumentary structures in these animals; however, this remains a task yet to be completed. Additionally, given that only exceptionally preserved fossils from Permo-Carboniferous lagerstätten with unique preservation conditions retain evidence of *in situ* delicate integumentary structures, a rigorous analysis of the influence of taphonomy on scale composition and histology is required to confirm their homology.

## 5.2. Evolution of integumentary patterns in Recumbirostra

The abundance of scales preserved on 'microsaur' fossils led Carroll & Gaskill [26] to comprehensively review the scale morphology of these animals. Generally, it seems likely that most 'microsaurs' were completely covered in dermal scales, including the dorsal, flank and ventral regions, as well as the appendages (limbs and tail) (figure 5). Most scales that have been observed in 'microsaurs' bear some variation of ridge patterning, if not other ultrastructural details (e.g. pits, secondary ridges, tubercles), that are only visible with microscopy. Despite this, Carroll & Gaskill [26] hypothesized that most 'microsaur' scales were smooth and that the ridge patterns found on scales are internal structures that were only superficially exposed by either taphonomic forces, or wear.

Careful examination of the scales of both recumbirostrans and other 'microsaurs' reveals this is certainly not the case—the scales on *Joermungandr* are just one example to the contrary.

Ornamented dermal scales are also well represented on specimens of *Microbrachis*, *Hyloplesion* (at present including FMNH PR 981), *Sparodus*, *Crinodon*, *Pelodosotis*, *Llistrofus* (see [49]), among others. Comparatively, only a few temnospondyl groups have convergently evolved similarly ornate scales to those of 'microsaurs'. Anastomosing longitudinal striae can be observed on micromelerpetids and some amphibamiforms [16,35]. Among fossorially adapted recumbirostrans, such as *Joermungandr*, it is possible that the distinct ridge patterns and other ultrastructural details are adaptations to burrowing where they would help shed substrate, similar to what is seen in extant fossorial reptile scales [46].

Carroll & Gaskill [26] noted that there are probably systematic differences in scale morphology across the many microsaurian families. However, at the time, the infrequent occurrence of scales allowed only

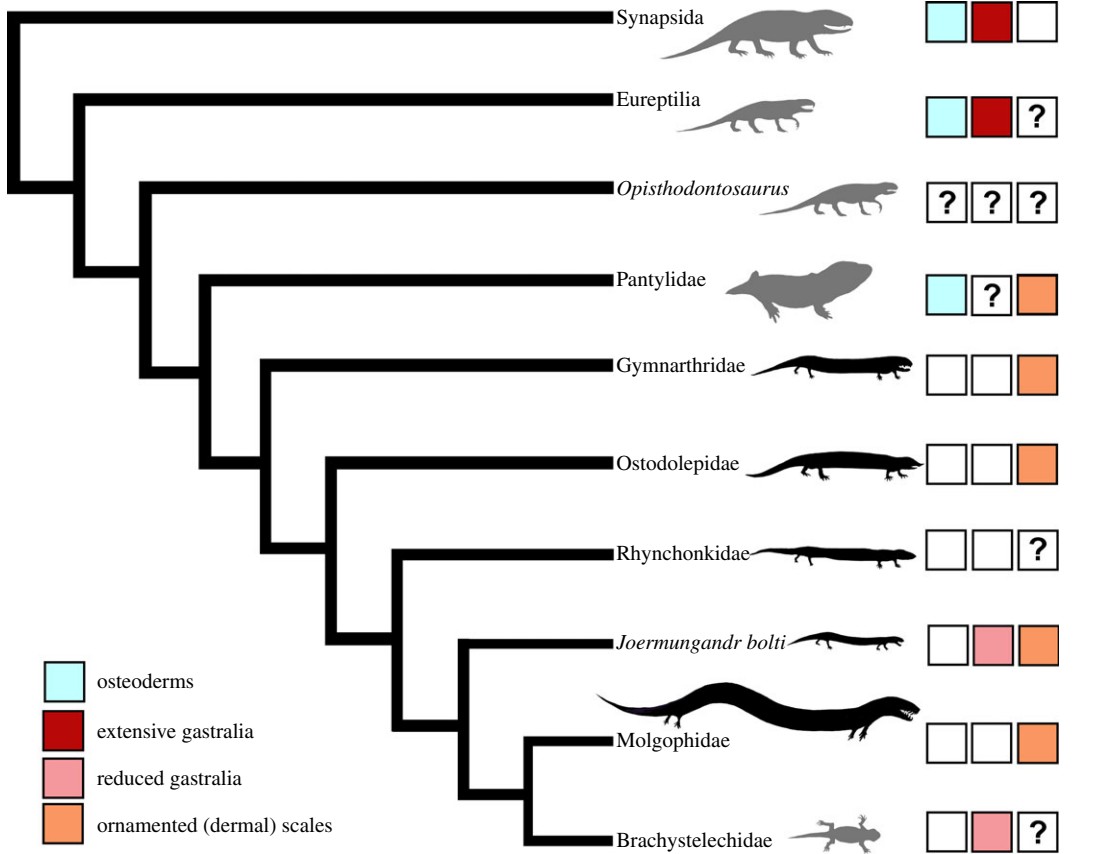

**Figure 5.** Simplified phylogeny of recumbirostrans showing integumentary structures and body elongation. Body elongation is indicated in silhouettes with grey = no body elongation; black = body elongation present. Question marks in boxes indicate uncertain character states rather than absences. Silhouettes for Ostodolepidae, Gymnarthridae, Rhynchonkidae and Synapsida modified from Pardo *et al*. [2].

for morphological descriptions and not for assessment of evolutionary patterns. What morphological patterns of scalature that can now be summarized for Recumbirostra are discussed here and reveal some clade level distinctions (figure 5). Early diverging, more generalized recumbirostrans such as *Steenerpeton* (formerly *Asaphestera intermedia*, see [6]) bear large, smooth, plate-like ovular scales that sometimes show concentric rings (possibly growth rings). This morphology is rare among recumbirostrans and may represent the plesiomorphic condition. Hapsidopariids, which appear to occupy a basal position in recumbirostran phylogeny [15,49], have scales known from *Hapsidoparieon*, *Llistrofus* and *Saxonerpeton*. The ovular scales of *Llistrofus* (see [49]) are highly ornamented, bearing ridges that form a radiating, clam shell-like pattern, sometimes with a bony webbing connecting ridges. The scales of *Llistrofus* can also bear a thicker raised margin on either of the anterior or posterior articulating ends. The scales of *Crinodon* appear morphologically similar to *Llistrofus*, however; placement of this 'microsaur' is currently uncertain. Ostodolepids show interesting variations in scale morphology with very fine, 'matted' scales present on *Pelodosotis* (see fig. 130C in Carroll & Gaskill [26]), whereas *Micraroter* (BPI 3839) bears very thick dermal scales that are even thicker on the ventral parts of the animal. Ornamentation may be present on these scales; however, they are not preserved well enough to parse out any details of the pattern.

Among *Joermungandr* and the clades Brachystelechidae and Molgophidae, there appear to be similar types of dermal scales that are very thin with ridged ornamentation. Molgophids appear to also bear thin dermal scales, but the details of these are not well preserved on any known specimen [28]. Of the Pantylidae, *Sparodus* has the best examples of an ornate-scale morphology consisting of fine ridges and perpendicular, evenly spaced, concentric circles [26]. No specimen of *Pantylus* shows dermal scalation, save one specimen (FMNH UR 1069) that only bears fragments. However, given this specimen and that the other members of this clade possess scales, it is likely *Pantylus* did as well. *Trachystegos* is heavily scaled, although its referral to Pantylidae is at present questionable. Scales are

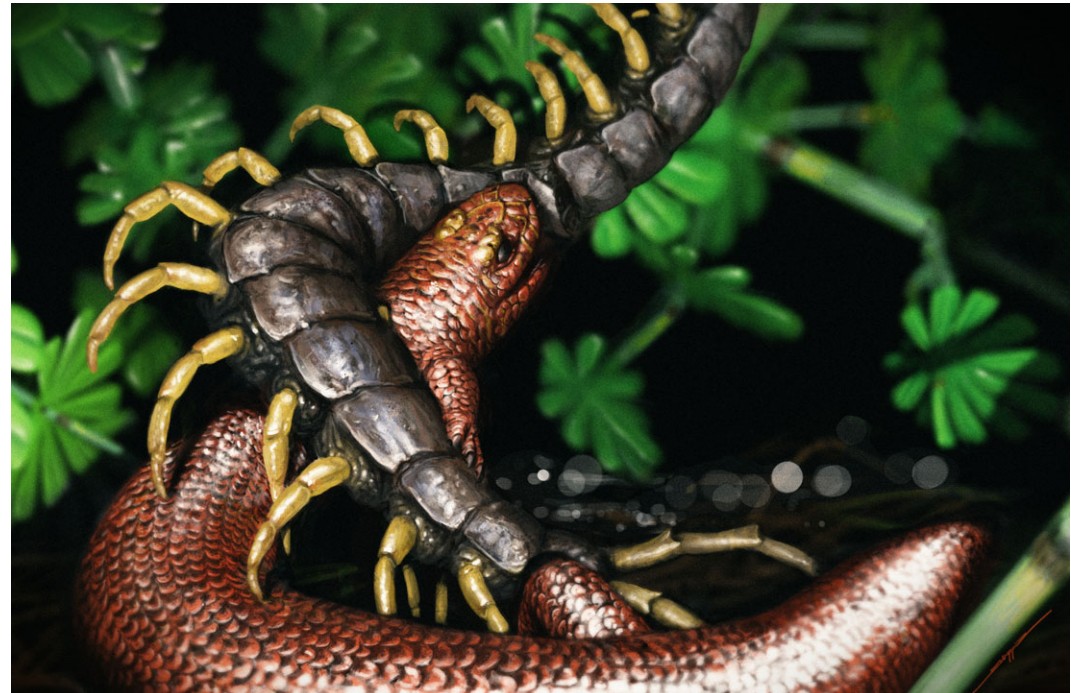

**Figure 6.** Artistic reconstruction of *J. bolti* gen. et sp. nov. battling with a centipede in the foliage of Mazon Creek. (Created by Henry Sutherland Sharpe, © 2019 Henry Sutherland Sharpe. Used under licence.)

not known in any member of the Rhynkonkidae, or curiously any member of the Gymnarthridae [26]. Although scales are often found preserved alongside gymnarthrids at Joggins, their association with a gymnarthrid is not certain due to the fragmentary nature of 'microsaur' material at the site and the presence of multiple tetrapods within a single stump. Recently, disarticulated scales were reported by Gee *et al*. [33] preserved alongside cranial remains of the gymnarthrid *Euryodus*.

The morphology of both dorsal and ventral scales is identical to one another in *Joermungandr*. Some recumbirostrans, however, show considerable variation in dermal scalation across the body, typically between the dorsal and ventral scales. For example, in *Micraroter*, the ventral scales are considerably thickened in comparison to the dorsal scales [26]. Ventral scales in the aforementioned recumbirostrans are still dermal scale in origin and not homologous with gastralia. That being said, there are a few cases where amniote-like gastralia are also found in recumbirostrans. The early recumbirostran *Steenerpeton* appears to bear a scattering of cylindrical gastralia [6]. Similarly, gastralia are also present in articulation on *Joermungandr* and short-bodied brachystelechids, e.g. *Diabloroter* and *Batropetes*, where they appear as thin, cylindrical structures, tapering on either side, that are morphologically similar to the gastralia of basal amniotes [14]. However, the gastralia of these recumbirostrans are markedly reduced, forming sparse, string-like, rows of ossifications that occupy a smaller surface area on the ventrum. This contrasts the chevrons of gastralia present in early amniotes such as *Cephalerpeton* [15,50,51], that form a continuous ventral matting of bone.

Given recent phylogenetic hypotheses that recumbirostrans are in fact amniotes descended from a captorhinid-like ancestor [2], it is possible that recumbirostrans at some point had true gastralia (figures 5 and 6). In this scenario, basal recumbirostrans such as *Steenerpeton* would retain the plesiomorphic condition for Recumbirostra, whereas *Joermungandr* and brachystelechids reveal a reversal possibly associated with unique ecologies (figure 5; [14]). This starkly contrasts the hypothesis of Carroll & Gaskill [26] who considered the order Microsauria to be stem-amniotes (Lepospondyli). In this scenario, their morphology was considered as derived from a basal tetrapod antecedent.

In addition to dermal scales and gastralia, Carroll & Gaskill [26] noted the presence of another kind of integumentary structure, bony 'ossicles' (a term used for small osteoderms), along the ventral regions of certain shorter-bodied 'microsaurs' (figure 5). *Pantylus*, *Stegotretus* and *Saxonerpeton* all bear unique hexagon-shaped osteoderms that line the ventral pectoral region of the animal (figure 5). Personal examination of these ossicles on *Pantylus*, *Stegotretus* and an unnamed pantylid from Nova Scotia reveal a dense assembly of these osteoderms that form a mosaic within the inter-dentary space extending posterior to the pectoral girdle (when preserved). These ossicles bear resemblance to the inter-dentary

osteoderms found on *Tuditanus*, *Crinodon* and *Cardiocephalus* and are probably homologous structures [26]. Carroll [52] proposed these osteoderms were derived from the ventral gastralia; however, we note that these ventral scales are not morphologically (small ossicles in a mosaic) or positionally (pectoral to inter-dentary space) similar to ventral gastralia, and are probably instead independently derived ossifications.

Unlike other Permo-Carboniferous groups such as dissorophids and chroniosuchians [35], recumbirostrans do not possess 'bulky' or large osteoderms. This is probably due to the functional constraints associated with a fossorial lifestyle, where large osteoderms protruding from the dermis would hinder both the locomotion and flexibility needed to achieve burrowing. Instead, the bodies of recumbirostrans such as *Joermungandr* appear from body outline impressions to have been streamlined, cylindrical and relatively smooth. Extant fossorial reptiles also have streamlined bodies that appear relatively smooth (e.g. amphisbaenians), lacking large osteoderms or other protruding keratinous scales or structures [46].

Finally, body elongation associated with increased fossoriality is common among recumbirostrans (only absent in brachystelechids and basal recumbirostrans) and even reaches extreme lengths in molgophids such as *Brachydectes* (up to 99 presacral vertebrae). However, similar to the lack of osteoderms or 'bony-ossicles' in longer-bodied recumbirostrans such as *Joermungandr* and molgophids, there is also a reduction and in some cases complete absence of gastralia (figure 5). Long-bodied recumbirostrans such as *Joermungandr* (figure 6) probably relied on lateral undulation or maybe some form of sidewinding [53,54] as a locomotory mode, and a reduction of ventral ossifications including the gastralia may have aided in providing flexibility to the ventrum (figure 5).

Ethics. No ethics assessment was required prior to the completion of this research because this study relied entirely on museum collections. Similarly, collecting permits were not required because no field collections were made.

Data accessibility. Additional supporting data are included in the electronic supplementary information. In addition, the nexus file used in the phylogenetic analysis has been deposited in the Dryad Digital Repository: https://doi.org/10.5061/dryad.63xsj3v1s [55].

The data are provided in electronic supplementary material [56]. This published work and the nomenclatural acts it contains have been registered in ZooBank: LSIDurn:lsid:zoobank.org:pub:75B96807-D3DB-466F-AF0E-A5C0BF64585D.

Authors' contributions. A.M. designed the study; A.M. and A.S.C. collected and analysed the data; A.M. and H.C.M. wrote the paper.

Competing interests. All authors declare no competing interests.

Funding. This project was partially funded by an Ontario Graduate Scholarship grant awarded to A.M.

Acknowledgements. First and foremost, we thank and are indebted to the late John Bolt for his kindness and generosity with sharing his knowledge and material. We thank William Simpson, Adrienne Stroup, Diane Scott, Robert Reisz, Dave Berman, Amy Henrici and Robert Hook for access to comparative material and providing helpful discussion on Permo-Carboniferous tetrapods. We further thank Jason Pardo and Emily McDaniel for providing discussion, support and help with figure construction. A special thanks to Henry Sharpe for his artistic reconstruction (painting) of *Joermungandr bolti* (figure 6). Finally, we thank Jason Anderson, Bryan Gee and another anonymous reviewer for helpful comments on earlier drafts of this manuscript.

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
