## [Peer Review File · Royal Society Open Science]

Review History

RSOS-201280.R0 (Original submission)

Review form: Reviewer 1 (Bryan Gee)

Is the manuscript scientifically sound in its present form?

Yes

Are the interpretations and conclusions justified by the results?

Yes

Is the language acceptable?

Yes

Do you have any ethical concerns with this paper?

No

Have you any concerns about statistical analyses in this paper?

No

Recommendation?

Accept with minor revision (please list in comments)

Comments to the Author(s)

To the authors,

I have reviewed the following manuscript describing a new recumbirostran 'microsaur' from Mazon Creek, the latest in a renewed interest in these early tetrapods. The manuscript is well-written and scientifically sound and should be acceptable for publication pending minor revisions. I have attached my detailed comments in the following document (see Appendix A).

Regards,
Bryan Gee

Review form: Reviewer 2 (Jason Anderson)

Is the manuscript scientifically sound in its present form?

Yes

Are the interpretations and conclusions justified by the results?

Yes

Is the language acceptable?

Yes

Do you have any ethical concerns with this paper?

No

Have you any concerns about statistical analyses in this paper?

No

Recommendation?

Major revision is needed (please make suggestions in comments)

Comments to the Author(s)

This is an interesting new tetrapod described from the famous Mazon Creek locality in Illinois, and includes a useful discussion of integumentary structures among. I support the eventual publication of this paper but there are a few changes that must take place before that can happen.

The largest issue is the proposed name itself. Article 27 of the ICZN expressly forbids the use of diacritical marks. The proposed name should thus remove the marks above the o, so the name would become Jormungandr or Joermungandr (which is how one represents the modern Swedish phoneme--I've no idea what the sound would be in Old Norse) or alternatively Jormungand (a common Anglicization). However the authors decide to go I would urge latinizing the name but I don't believe the ICZN mandates this (as long as the language of origin is specified). This change is absolutely mandatory, which is why I gave the paper "major revisions" instead of the "minor revisions" rating it otherwise deserves.

Also mandatory is that the authors must specify gender of the generic name. Modern Swedish does not have male/female genders but Old Norse did as I understand it; regardless this is a proper name that is being proposed, so double check the ICZN to be sure there aren't any specific regulations in this regard and specify your preference explicitly in the Systematic section..

(A more pedantic quibble: the authors also should be cautious when they mention Jörmungandr being the "world serpent" (which I accept is a thing it is called). Jörmungandr is the serpent that dwells in the Midgard sea; Midgard is only one of several worlds in Norse mythology. Nidhoggr is the serpent that feeds on the roots of the World Tree Yggdrasil, which spans all of these other "worlds" (or realms) and is what first jumped to my mind since "world" can be interpreted as "cosmos" in this context. A little more clarification would help avoid others making the same misunderstanding.)

Specific comments:

PDF page 4: citation in abstract

page 5: the plural of bauplan is baupläne

page 9 line 3: these features mentioned setting Odonterpeton aside are all likely ontogenetic.

line 34: presumably you mean Quasicaecilia not Cardiocephalus (which has an elongate, bullet-shaped head)?

page 10 line 21: the font size changes here (and elsewhere in the MS)

page 14 line 20: rethink the choice of "worthy", as its not a question of merit but relevance

page 17 line 54: replace "revamped" with "recent"

Figure 5 was too large to download and view in the PDF, oddly since its just a cladogram and can be presented as a greyscale image, so I was not able to evaluate it.

Interesting stuff, I look forward to these mandatory changes being made and consideration of the other points in a future revised MS or the final published paper!

Jason Anderson

Review form: Reviewer 3

Is the manuscript scientifically sound in its present form?

No

Are the interpretations and conclusions justified by the results?

Yes

Is the language acceptable?

Yes

Do you have any ethical concerns with this paper?

No

Have you any concerns about statistical analyses in this paper?

No

Recommendation?

Major revision is needed (please make suggestions in comments)

Comments to the Author(s)

I support the publication of this manuscript, but feel that quite a lot of additional work is required. Some aspects of the manuscript are in conflict with the illustrations and the available data. For example the authors talk about paired parietals and frontals, but only single elements are shown in the figure. Is the element to the left of the bone identified as frontal actually the left frontal or another element? Possibly, but the skull as preserved is incomplete and it is not clear. The authors refer to a pineal foramen, but this is again not shown in the illustration. The photographs in Figure 3 are not clear, therefore the interpretations of the cranial anatomy are in doubt. Figure 3B does not show clearly that the element identified as frontal is excluded from the orbit.

I also have trouble with the identification of a short tail. The last caudal is at the tip of the concretion, and there is no evidence that this is the end of the tail.

Lastly, the information on the scales is very interesting and highly informative, but the absence of gastralia is uncertain. I am not familiar with the condition of this kind material, but it appears to be quite 2D, and flattened with only a small part of the outer surface of the fossil being preserved, and when the concretion broke only some of the fossil remains were exposed. In fact, as described and shown, only a small surface of the overall fossil shows the presence of scales. Thus, it is not possible to determine, in my opinion, if gastralia were present or not. I think this is a major weakness in the paper because of the issue of homology of gastralia and the scales, and their relevance to the ecomorphology of the taxon. The description of the scales states that dorsal, lateral, and ventral scales are preserved, but no real evidence of this is provided. It appears as if only the part of the right side of the body fossil shows scales. It is highly unlikely that these impressions of scales represent the dorsal, flank, and ventral regions of the body, and simply stating that without any evidence is not acceptable. I would urge you to reconsider this part of the manuscript and address these concerns.

Decision letter (RSOS-201280.R0)

Dear Mr Mann

The Editors assigned to your paper RSOS-201280 "Jörmungandr bolti, an exceptionally preserved 'microsaur' from Mazon Creek, Illinois, reveals patterns of integumentary evolution in Recumbirostra." have made a decision based on their reading of the paper and any comments received from reviewers.

Regrettably, in view of the reports received, the manuscript has been rejected in its current form. However, a new manuscript may be submitted which takes into consideration these comments.

We invite you to respond to the comments supplied below and prepare a resubmission of your manuscript. Below the referees' and Editors' comments (where applicable) we provide additional requirements. We provide guidance below to help you prepare your revision.

Please note that resubmitting your manuscript does not guarantee eventual acceptance, and we do not generally allow multiple rounds of revision and resubmission, so we urge you to make every effort to fully address all of the comments at this stage. If deemed necessary by the Editors, your manuscript will be sent back to one or more of the original reviewers for assessment. If the original reviewers are not available, we may invite new reviewers.

Please resubmit your revised manuscript and required files (see below) no later than 05-Apr-2021. Note: the ScholarOne system will 'lock' if resubmission is attempted on or after this deadline. If you do not think you will be able to meet this deadline, please contact the editorial office immediately.

Please note article processing charges apply to papers accepted for publication in Royal Society Open Science (<https://royalsocietypublishing.org/rsos/charges>). Charges will also apply to papers transferred to the journal from other Royal Society Publishing journals, as well as papers submitted as part of our collaboration with the Royal Society of Chemistry (<https://royalsocietypublishing.org/rsos/chemistry>). Fee waivers are available but must be requested when you submit your manuscript (<https://royalsocietypublishing.org/rsos/waivers>).

Thank you for submitting your manuscript to Royal Society Open Science and we look forward to receiving your resubmission. If you have any questions at all, please do not hesitate to get in touch.

on behalf of Prof Kevin Padian (Subject Editor)
openscience@royalsociety.org

Editor comments:

Dear Authors: thanks for your submission. As you can see the reviewers are generally positive about the work but each of them has substantial concerns; and, interestingly enough, they are all pretty different from each other. On balance I am giving a "reject/resub" decision mainly because you will have more time to deal with their concerns, not the least of which is the illustrations. Please prepare a response to each concern with your resubmission; we normally send the resubs out for another round of review and so it will be important to meet all the concerns of the reviewers. Best wishes.

Reviewer comments to Author:

Reviewer: 1
Comments to the Author(s)
To the authors,

I have reviewed the following manuscript describing a new recumbirostran 'microsaur' from Mazon Creek, the latest in a renewed interest in these early tetrapods. The manuscript is well-written and scientifically sound and should be acceptable for publication pending minor revisions. I have attached my detailed comments in the following document (in both PDF and Word formats).

Regards,
Bryan Gee

Reviewer: 2

Comments to the Author(s)

This is an interesting new tetrapod described from the famous Mazon Creek locality in Illinois, and includes a useful discussion of integumentary structures among. I support the eventual publication of this paper but there are a few changes that must take place before that can happen.

The largest issue is the proposed name itself. Article 27 of the ICZN expressly forbids the use of diacritical marks. The proposed name should thus remove the marks above the o, so the name would become Jormungandr or Joermungandr (which is how one represents the modern Swedish phoneme--I've no idea what the sound would be in Old Norse) or alternatively Jormungand (a common Anglicization). However the authors decide to go I would urge latinizing the name but I don't believe the ICZN mandates this (as long as the language of origin is specified). This change is absolutely mandatory, which is why I gave the paper "major revisions" instead of the "minor revisions" rating it otherwise deserves.

Also mandatory is that the authors must specify gender of the generic name. Modern Swedish does not have male/female genders but Old Norse did as I understand it; regardless this is a proper name that is being proposed, so double check the ICZN to be sure there aren't any specific regulations in this regard and specify your preference explicitly in the Systematic section..

(A more pedantic quibble: the authors also should be cautious when they mention Jörmungandr being the "world serpent" (which I accept is a thing it is called). Jörmungandr is the serpent that dwells in the Midgard sea; Midgard is only one of several worlds in Norse mythology. Nidhoggr is the serpent that feeds on the roots of the World Tree Yggdrasil, which spans all of these other "worlds" (or realms) and is what first jumped to my mind since "world" can be interpreted as "cosmos" in this context. A little more clarification would help avoid others making the same misunderstanding.)

Specific comments:

PDF page 4: citation in abstract

page 5: the plural of bauplan is baupläne

page 9 line 3: these features mentioned setting Odonterpeton aside are all likely ontogenetic.

line 34: presumably you mean Quasicaecilia not Cardiocephalus (which has an elongate, bullet-shaped head)?

page 10 line 21: the font size changes here (and elsewhere in the MS)

page 14 line 20: rethink the choice of "worthy", as its not a question of merit but relevance

page 17 line 54: replace "revamped" with "recent"

Figure 5 was too large to download and view in the PDF, oddly since its just a cladogram and can be presented as a greyscale image, so I was not able to evaluate it.

Interesting stuff, I look forward to these mandatory changes being made and consideration of the other points in a future revised MS or the final published paper!

Jason Anderson

Reviewer: 3

Comments to the Author(s)

I support the publication of this manuscript, but feel that quite a lot of additional work is required. Some aspects of the manuscript are in conflict with the illustrations and the available data. For example the authors talk about paired parietals and frontals, but only single elements

are shown in the figure. Is the element to the left of the bone identified as frontal actually the left frontal or another element? Possibly, but the skull as preserved is incomplete and it is not clear. The authors refer to a pineal foramen, but this is again not shown in the illustration. The photographs in Figure 3 are not clear, therefore the interpretations of the cranial anatomy are in doubt. Figure 3B does not show clearly that the element identified as frontal is excluded from the orbit.

I also have trouble with the identification of a short tail. The last caudal is at the tip of the concretion, and there is no evidence that this is the end of the tail.

Lastly, the information on the scales is very interesting and highly informative, but the absence of gastralia is uncertain. I am not familiar with the condition of this kind material, but it appears to be quite 2D, and flattened with only a small part of the outer surface of the fossil being preserved, and when the concretion broke only some of the fossil remains were exposed. In fact, as described and shown, only a small surface of the overall fossil shows the presence of scales. Thus, it is not possible to determine, in my opinion, if gastralia were present or not. I think this is a major weakness in the paper because of the issue of homology of gastralia and the scales, and their relevance to the ecomorphology of the taxon. The description of the scales states that dorsal, lateral, and ventral scales are preserved, but no real evidence of this is provided. It appears as if only the part of the right side of the body fossil shows scales. It is highly unlikely that these impressions of scales represent the dorsal, flank, and ventral regions of the body, and simply stating that without any evidence is not acceptable. I would urge you to reconsider this part of the manuscript and address these concerns.

===PREPARING YOUR MANUSCRIPT===

===PREPARING YOUR REVISION IN SCHOLARONE===

Author's Response to Decision Letter for (RSOS-201280.R0)

See Appendix B.

RSOS-210319.R0

Review form: Reviewer 1 (Bryan Gee)

Is the manuscript scientifically sound in its present form?

Yes

Are the interpretations and conclusions justified by the results?

Yes

Is the language acceptable?

Yes

Do you have any ethical concerns with this paper?

No

Have you any concerns about statistical analyses in this paper?

No

Recommendation?

Accept as is

Comments to the Author(s)

Thank you for resubmitting your revised manuscript; I am pleased to see that the requested changes have been incorporated into the text. I noted a few errors or areas where the wording could be improved below, but in my opinion, the manuscript is now acceptable for publication (see Appendix C).

Review form: Reviewer 2 (Jason Anderson)

Is the manuscript scientifically sound in its present form?

Yes

Are the interpretations and conclusions justified by the results?

Yes

Is the language acceptable?

Yes

Do you have any ethical concerns with this paper?

No

Have you any concerns about statistical analyses in this paper?

No

Recommendation?

Accept with minor revision (please list in comments)

Comments to the Author(s)

I thank the authors for accommodating my earlier comments into the present revision, resulting in a much improved MS. There are still some minor revisions needed; I have made numerous suggestions in the attached PDF (see Appendix D). The phylogenetic analysis for some reason precedes the description, making for analysis for a taxon yet to be named! It should follow the syst paleo and description. Be sure that all abbreviations in the figures are defined in text (currently they are not).

My major scientific issue comes with the identification of the "supraoccipital". In the figure of the skull (which is slightly oblique of a full dorsal view) the midline suture appears to continue posterior of the parietals, and through the "supraoccipital", making it a paired structure, which would be odd indeed. I strongly encourage the authors to consider they instead have illustrated paired postparietals. If this interpretation is accepted then it will require some follow on alteration of the description and discussion and will probably impact upon their phylogenetic results.

Decision letter (RSOS-210319.R0)

Dear Mr Mann

On behalf of the Editors, we are pleased to inform you that your Manuscript RSOS-210319 "Joermungandr bolti, an exceptionally preserved 'microsaur' from Mazon Creek, Illinois, reveals patterns of integumentary evolution in Recumbirostra." has been accepted for publication in Royal Society Open Science subject to minor revision in accordance with the referees' reports. Please find the referees' comments along with any feedback from the Editors below my signature.

Please submit your revised manuscript and required files (see below) no later than 7 days from today's (ie 18-May-2021) date. Note: the ScholarOne system will 'lock' if submission of the revision is attempted 7 or more days after the deadline. If you do not think you will be able to meet this deadline please contact the editorial office immediately.

on behalf of Prof Kevin Padian (Subject Editor)
openscience@royalsociety.org

Editor comments:

Thanks for your revisions and congratulations. Please address the reviewer's comments specifically when you submit your final version.

Reviewer comments to Author:

Reviewer: 1

Comments to the Author(s)

Thank you for resubmitting your revised manuscript; I am pleased to see that the requested changes have been incorporated into the text. I noted a few errors or areas where the wording could be improved below, but in my opinion, the manuscript is now acceptable for publication.

Reviewer: 2

Comments to the Author(s)

I thank the authors for accommodating my earlier comments into the present revision, resulting in a much improved MS. There are still some minor revisions needed; I have made numerous suggestions in the attached PDF. The phylogenetic analysis for some reason precedes the description, making for analysis for a taxon yet to be named! It should follow the syst paleo and description. Be sure that all abbreviations in the figures are defined in text (currently they are not).

My major scientific issue comes with the identification of the "supraoccipital". In the figure of the skull (which is slightly oblique of a full dorsal view) the midline suture appears to continue posterior of the parietals, and through the "supraoccipital", making it a paired structure, which would be odd indeed. I strongly encourage the authors to consider they instead have illustrated paired postparietals. If this interpretation is accepted then it will require some follow on alteration of the description and discussion and will probably impact upon their phylogenetic results.

===PREPARING YOUR MANUSCRIPT===

===PREPARING YOUR REVISION IN SCHOLARONE===

- Any electronic supplementary material (ESM).
- If you are requesting a discretionary waiver for the article processing charge, the waiver form must be included at this step.
- If you are providing image files for potential cover images, please upload these at this step, and inform the editorial office you have done so. You must hold the copyright to any image provided.
- A copy of your point-by-point response to referees and Editors. This will expedite the preparation of your proof.

- Ensure that your data access statement meets the requirements at <https://royalsociety.org/journals/authors/author-guidelines/#data>. You should ensure that you cite the dataset in your reference list. If you have deposited data etc in the Dryad repository, please only include the 'For publication' link at this stage. You should remove the 'For review' link.
- If you are requesting an article processing charge waiver, you must select the relevant waiver option (if requesting a discretionary waiver, the form should have been uploaded at Step 3 'File upload' above).
- If you have uploaded ESM files, please ensure you follow the guidance at <https://royalsociety.org/journals/authors/author-guidelines/#supplementary-material> to include a suitable title and informative caption. An example of appropriate titling and captioning may be found at https://figshare.com/articles/Table_S2_from_Is_there_a_trade-off_between_peak_performance_and_performance_breadth_across_temperatures_for_aerobic_scope_in_teleost_fishes_/3843624.

Author's Response to Decision Letter for (RSOS-210319.R0)

See Appendix E.

Decision letter (RSOS-210319.R1)

Dear Mr Mann,

I am pleased to inform you that your manuscript entitled "Joermungandr bolti, an exceptionally preserved 'microsaur' from Mazon Creek, Illinois, reveals patterns of integumentary evolution in Recumbirostra." is now accepted for publication in Royal Society Open Science.

If you have not already done so, please remember to make any data sets or code libraries 'live' prior to publication, and update any links as needed when you receive a proof to check - for

instance, from a private 'for review' URL to a publicly accessible 'for publication' URL. It is good practice to also add data sets, code and other digital materials to your reference list.

You can expect to receive a proof of your article in the near future. Please contact the editorial office (openscience@royalsociety.org) and the production office (openscience_proofs@royalsociety.org) to let us know if you are likely to be away from e-mail contact – if you are going to be away, please nominate a co-author (if available) to manage the proofing process, and ensure they are copied into your email to the journal. Due to rapid publication and an extremely tight schedule, if comments are not received, your paper may experience a delay in publication.

on behalf of Professor Kevin Padian (Subject Editor)
openscience@royalsociety.org

Appendix A

Review of RSOS-201280 (Mann, Calthorpe & Maddin)

To the authors,

I have reviewed the following manuscript describing a new recumbirostran from Mazon Creek, the latest in a recent resurgence in the study of these important nodule-encapsulated tetrapods. The manuscript is well-presented and is important for further resolving the origins of various tetrapod clades, both within crown Amniota and beyond, and the present study contributes further to this endeavour. I recommend this manuscript for publication pending minor revisions, most of which are suggestions that I hope will improve the flow and clarity of the text for the potential readers who are not as regularly enthralled by the peculiar recumbirostrans and the nuances of early amniote evolution. As two of the authors are well aware, I tend to write rather verbosely, and the length of my review is not a direct comment on my opinion of the validity or import of the manuscript. My below comments are divided into general and specific line remarks. If the authors have any questions, they are welcome to contact me (via my new email: bmgee@uw.edu).

Best wishes,

Bryan Gee
Postdoctoral scholar
University of Washington

General comments

1. There are a number of instances where ‘microsaur’ is put in single quotation marks, as with most recent workers, but also instances where it is not (e.g., p. 14, l. 16, 28); this should be standardized in the revision since I doubt it will be during copy-editing.
2. There are numerous instances in which large summary works (particularly Carroll & Gaskill) are cited as the only reference for a specific statement or figure. Especially for directing readers to figures that are not reproduced here (the scales of *Hyloplesion* is a good example to permit “direct comparison” by the reader), I would suggest adding the specific figure or pagination, as the authors did for *Pelodosotis* scales.
3. The author contributions as entered in the submission portal differ slightly from those listed in the manuscript; the authors should just double check for consistency / their preferred version.
4. As the most recent editor of the utilized matrix, OMNH 53519 is *Euryodus* sp. from Richards Spur, FMNH UR 2296 is the holotype of *Euryodus dalyae*, and ‘*Euryodus dalyae* (composite)’ is FMNH UR 2296 + additional isolated referred material listed by Carroll & Gaskill (1978). I had scored the OTUs this way to test the affinities of the Richards Spur material, but don’t run all three OTUs together. I doubt it will change the topology, but this is a minor detail that needs to be addressed.
5. The authors should make sure to double-check the italicization of scientific names in their references because this is something often missed or erroneously altered during copy-editing and typesetting.

Review submitted: September 20, 2020

6. I made a few specific comments (e.g., lacrimal-jugal contact, pineal-frontal contact), but there are a few areas where the anatomy as illustrated seems to suggest taphonomic distortion other than just the folding of one side of the skull. It would be helpful for future workers who cannot examine the specimen first-hand to have a more explicit description of the entire physical state of the specimen with this in mind. This is also true of differentiating when something is ‘not exposed’ versus ‘not preserved’ in the text.

Specific line comments

Note: pagination refers to the pagination of the submitted manuscript file, rather than that of the reviewing PDF

- P. 1, l. 27: perhaps consider a running header that gives some indication of what general group this organism belongs to, especially because this is a general science journal.
- P. 2, abstract: the content is appropriate, but perhaps it would be better to consolidate the sentences discussing integumentary ossifications together. Right now, it is a little disjointed in mentioning scale ultrastructure first, then recumbirostran buaplan, then phylogenetic analyses, and then ending on the note of reviewing integumentary features after already discussing the scales.
- P. 2, l. 47-51: this sentence is a little confusing because 315 Ma is not the middle of the Carboniferous; if the authors are indicating molecular clock calculations that imply an older origin more around 325–330 Ma that pre-dates the fossil record, they should provide citations for that.
- P. 3, l. 5: which group (or both groups)?
- P. 3, l. 10: do you mean ‘recumbirostran amniote’ (implying they are crown amniotes) or ‘recumbirostran-amniote’ (implying they are distinct from [crown] amniotes and part of a bracket for some faunal biozone / assemblage, like the S. African AZs or the Russian Triassic stages, for example).
- P. 3, l. 31-33: this sentence reads a little awkwardly, perhaps move the clause after the comma to after “includes.”
- P. 4, l. 24: institutional abbreviation for the Canadian Museum of Nature?
- P. 4, l. 24-26: you might want to specify that it is only a table listing what extant specimens you studied is available in supplemental. The present wording implies that the supplement includes a slew of comparative high-magnification and SEM images of extant reptile scales (I myself was expecting this). In the same vein a few lines up, I don’t know that you can say that you “compiled a dataset” because the table is not a dataset in the sense that anyone else can really use it for other studies; it is more of a ‘materials used/examined.’
- P. 4, l. 35: aren’t all vertebrae ‘dorsal’ in position? I would just simplify it to ‘v=vertebrae’ since you don’t use ‘v’ for any other element
- P. 5, l. 19-21: how many bootstrap replicates were run?
- P. 27, l. 27-29: the depiction of characters and states here is a little odd; writing it as 13(1) is not only more common but also easier to read when listing several characters.
- P. 5, l. 31: I am pretty sure that referencing the phylogeny figure here would require that it become Figure 1 and featured first in the published version because no other figure was

mentioned prior to this; so you might consider omitting this sentence and putting it in the figure caption to keep the order of figures.

- P. 5, l. 51: is the latex peel catalogued under the same number (referring to p. 4, l. 7-9)? This is also not clear in the supplemental caption.
- P. 6, l. 21: perhaps consider expanding a little on the ‘short tooth rows’ because it could be variably construed to mean ‘short jaw in general’ or ‘unusually short pre-coronoid-process part of the jaw’ in addition to the condition here (a normal proportioned jaw with the tooth row terminating well anterior to the normal position)
- P. 6, l. 25: I would specify that you mean the co-ossification of the otic capsules and the occiput since ‘braincase’ can be variably interpreted (and sometimes includes the supraoccipital).
- P. 6, l. 33: I suggest adding an opening sentence or two that introduces *Odonterpeton* a little more broadly for less familiar readers; this is the first mention of the taxon (which is not from Mazon Creek), and it is not tested in this matrix, so it’s not immediately apparent why you draw these detailed comparisons.
- P. 6, l. 35: ‘Microbrachomorpha’
- P. 7, l. 28: what do you mean by ‘robust?’ Usually this refers to the degree of ossification, but it seems that perhaps you are referring more to the short and blunt skull profile?
- P. 7, l. 30-34: this seems like a strange comparison because the skull of *Cardiocephalus* is much longer than that of *Carrolla* and without a markedly blunt snout; *Carrolla* is more equilaterally triangular and blunted like *Jörmungandr*.
- P. 7, l. 38: ‘preserved’ or ‘exposed?’
- P. 7, l. 52: how confident are the authors that the frontal is totally excluded? While the postfrontal process is long, the preserved configuration suggests that a very small part of the frontal could enter the orbit.
- P. 8, l. 15: is that a ramus or just the main body of the postfrontal?
- P. 8, l. 15-20: in *Diabloroter*, the pineal foramen is near the parietal-frontal suture, whereas here it contacts the frontal. The authors noted that the skull has partially folded, but has there been any dislodging and anteroposterior shifting of the elements?
- P. 8, l. 22: this characterization is a little confusing because it’s not clear whether all of the elements were fully dorsally exposed (i.e. which elements contribute to the occipital margin) and because ‘rounded’ is a little generic for describing that margin (‘undulating’ is how it’s scored, for example).
- P. 8, l. 48: I don’t doubt that the jugal contacts the maxilla anteroventrally, but is this contact preserved on either side? If not, you should only say that it can be inferred.
- P. 9, l. 7: the wording here is a little confusing because “processes [plural] on the prefrontals” could be interpreted to mean multiple processes on each prefrontal (and the figure of the left one can have this effect).
- P. 9, l. 20: more specifically just below the anterior orbital margin? “Just below the orbit” could be anywhere along the orbit.
- P. 9, l. 32-34: this is a little confusing – by ‘left lateral view,’ do you mean when viewed from above (part B) or just that the left jaw is exposed laterally? Isn’t the laterally exposed jaw the left jaw (contrary to the next sentence where it says the left jaw is exposed ventrally)?

- P. 10, l. 10: is the cultriform process fully exposed? If so, that is remarkably short relative to the body of the parasphenoid (and probably apomorphic as a result); it also looks proportionately much shorter in Fig. 3D than it does in the simplified Fig. 2B.
- P. 10, l. 44: the text says “unidentified,” but the figure has it labeled as the ulna?
- P. 11, l. 12: can scales really be considered “soft tissue anatomy” given the talking points about the tissue structure in the Discussion? They might be embedded in the dermis, but the main skeleton is also “embedded” in soft tissue. Perhaps it would be better to say ‘rarely preserved features, including soft body impressions...’
- P. 11, l. 21: what reference taxa is the “long limbs” characterization being implicitly compared to?
- P. 11, l. 46: what is meant by ‘cranial scales?’ This is the first mention of these scales, and it’s not apparent where they fall in the dorsal vs. ventral vs. flank categories (and also if they are on the head, why are they not figured in Fig. 3, and if not, how are you sure that they are cranial scales).
- P. 12, l. 7: sounds a little awkward ending on ‘between.’
- P. 12, l. 45-47: I suggest specifying that those workers focused on the nomenclatural divisions primarily in early tetrapods if only because they were not directly addressing homology for all tetrapods.
- P. 13, l. 17-19: aberrant as in having strange microstructure compared with that of other recumbirostrans or aberrant in having microstructure to begin with?
- P. 13, l. 32: ‘recumbirostran’
- P. 14, l. 17: ‘micromelerpetids’ and ‘amphibamiforms’
- P. 14, l. 38: it seems strange to call what is interpreted as the plesiomorphic condition ‘aberrant’; is ‘rare’ or ‘unusual’ a better term?
- P. 14, l. 40: ‘Hapsidopareiiids’ following Marjanovic & Laurin (2019).
- P. 14, l. 42: the initial derivations of the Pardo et al. matrix maintained their somewhat ambiguous position near the base of Recumbirostra, but the Gee et al. (2020) results, like those here, shift them higher as the sister group to ostodolepids; I suggest either switching the citation to my 2019 paper or making a note of the uncertainty around hapsidopareiiids.
- P. 15, l. 7: you might need to explicitly describe what this group is because it is so rarely used or mapped on phylogenies and because historically any derivation from *Cocytinus* has been suppressed in favour of other variants (e.g., Pardo & Anderson, 2016).
- P. 15, l. 30: see Supp. Fig. 6 from my *Euryodus* paper, which is a close-up of scales on the skull roof of one of the Richards Spur specimens. You might consider caveating the statement that conditions were not conducive for preservation of scales, etc. for gymnarthrids since Richards Spur does often preserve a lot of small structures (and did for *Euryodus* sp. and *Llistrofus* from the site).
- P. 16, l. 3: based on the concluding sentences, it sounds like you are proposing that specifically early recumbirostrans had gastralia and then lost them as they progressed to a more fossorial lifestyle; I would say this rather than the vague ‘at some point.’
- P. 16, l. 12: by ‘their morphology,’ do you mean of the scales specifically or the entire skeleton?
- P. 16, l. 36-41: I would mention chroniosuchians as well, specifically because they also have a similar mode of articulation with the vertebrae for stiffening the backbone to

dissorophids (an equally important aspect barring fossoriality); dissorophids alone is an odd comparison. It might also be a good idea to point out suggestions of temnospondyls burrowing are very rare and not inferred based on scale / osteoderm morphology; many other taxa have coverings of dermal ossifications (e.g., *Ecolsonia*, *Laidleria*, plagiosaurids) and are pretty confidently not primary burrowers.

- P. 16, l. 52: what are pantylids if not basal recumbirostrans?

Figures

- Figure 1
 - an alternative way to configure Figures 1 and 2 would be to pair each photograph with the corresponding line illustration so that it is easier to directly compare features (presumably these figures will be too large to be on the same page).
- Figure 2
 - I recommend increasing the size of the labeling text.
- Figure 3
 - perhaps ‘mirrored’ would be preferable to ‘reversed?’
 - capitalization of ‘Photoshop.’
 - in Part D, I think any label that is not within the element should have a line; it will just make it simpler for the reader and more internally consistent.
 - based on the description on p. 9-10, it seems like the element labeled “pl ?” is actually the possible pterygoid and the unlabeled element anterior to it is the palatine/ectopterygoid?
 - what is the element behind the left maxilla? Could that not be the quadratojugal?
- Figure 4
 - should there be a title before the subcaptions for the specific parts of the figure?
 - the text in parts A-C is a little hard to read; I would suggest either adding a stroke to it or a background box.
 - the text in part D is very small and should be increased in size.
- Figure 5
 - I recommend visually pruning the tree to focus only on ‘microsaurs’ or Amniota, which would allow you to make the important part of the tree visually larger; the addition of the new taxon has no measurable effect on temnospondyls and lissamphibians, for example, and there also appears to be no important changes outside of crown Tetrapoda.
- Figure 6
 - in my opinion, the lines are unnecessarily thick and dwarf the names of the clades, which are even smaller than the osteological features key.
 - italicization of genus/species names?
 - spelling of ‘Brachystelechidae’ and change ‘Eureptiles’ to ‘Eureptilia.’
 - I would consider putting a single node for Recumbirostra.
 - if you obtained any of these silhouettes from non-authors, I believe that needs to be indicated somewhere in the text.
- Figure 7
 - is this figure cited anywhere in the text?

Supplemental

- Spelling and capitalization of ‘arboreal’ in the supplemental table listing examined extant squamates.
- All of the extant taxa are squamates, so is there a reason why the “higher taxa” box cannot be consolidated to just the relevant lower taxonomic ranks?
- Character 14: coded as lacrimal-jugal contact ‘absent,’ but Figure 3 clearly shows contact on the right side. I don’t think they contacted when alive, but there is no discussion of deformation that artificially produced this contact in the text.
- Character 15: coded as quadratojugal ‘?,’ but as I noted above, what is the fragment behind the left maxilla if not the quadratojugal? This would affect #17 as well if interpreted as a quadratojugal.
- Character 40: at present, there is no description of an orbital rim, nor is one apparent in the figures.
- Character 68: coded as 20-29 maxillary teeth, but the description says 11-12.
- Character 101: I am not convinced that either mandible is sufficiently exposed to be able to confidently show that the splenial did not contribute to the symphysis. If it had a contribution, it would be on the lingual surface, which is not exposed on the left mandible, and no splenial or postsplenial is present / exposed on the right mandible.
- Character 138: the text says no atlas elements are identified, so I am wondering how the atlas ribs can be coded for ‘one pair.’
- Character 143: in a similar vein, if the pelvis is mostly absent, how confident is the determination of only a single pair of sacral ribs?
- Character 184: I think that this is a textbook example of anterior waisting (constriction) of the parietals.

Appendix B

Response to Reviewers Comments:

Editor comments:

Dear Authors: thanks for your submission. As you can see the reviewers are generally positive about the work but each of them has substantial concerns; and, interestingly enough, they are all pretty different from each other. On balance I am giving a "reject/resub" decision mainly because you will have more time to deal with their concerns, not the least of which is the illustrations. Please prepare a response to each concern with your resubmission; we normally send the resubs out for another round of review and so it will be important to meet all the concerns of the reviewers. Best wishes.

Response: Dear editor,

I have addressed the three reviewer's comments to the best of my ability. Virtually every concern of the first and second reviewer have been addressed, and significant changes have been made to the original submission that I hope are to your acceptance. Most of reviewer 3's comments were also addressed and significantly improved the manuscript, however, some of this reviewers' comments and concerns arose from unfamiliarity with the preservation of the material and have not been adopted.

Thanks.

Arjan Mann

See below my detailed responses:

Reviewer comments to Author:

Reviewer: 1

Comments to the Author(s)

To the authors,

I have reviewed the following manuscript describing a new recumbirostran 'microsaur' from Mazon Creek, the latest in a renewed interest in these early tetrapods. The manuscript is well-written and scientifically sound and should be acceptable for publication pending minor revisions. I have attached my detailed comments in the following document (in both PDF and Word formats).

Regards,
Bryan Gee

Response: Thanks for your detailed review, all of the minor revisions have been addressed in your attached Pdf, including most figure changes (save some that were just suggestions) as well as supplemental and coding changes as well. New additions include new phylogenetic analysis and discussion in the description on some tenuous characters. Major/general comments have been addressed, and all line changes have been accepted see below:

General comments

1. There are a number of instances where 'microsaur' is put in single quotation marks, as with most recent workers, but also instances where it is not (e.g., p. 14, l. 16, 28); this should be standardized in

the revision since I doubt it will be during copy-editing.

Response: changed.

2. There are numerous instances in which large summary works (particularly Carroll & Gaskill) are cited as the only reference for a specific statement or figure. Especially for directing readers to figures that are not reproduced here (the scales of *Hylopleston* is a good example to permit “direct comparison” by the reader), I would suggest adding the specific figure or pagination, as the authors did for *Pelodosotis* scales.

Response: done where applicable.

3. The author contributions as entered in the submission portal differ slightly from those listed in the manuscript; the authors should just double check for consistency / their preferred version.

Response: The manuscript is the preferred version and this will be changed accordingly

4. As the most recent editor of the utilized matrix, OMNH 53519 is *Euryodus* sp. from Richards Spur, FMNH UR 2296 is the holotype of *Euryodus dalyae*, and ‘*Euryodus dalyae* (composite)’ is FMNH UR 2296 + additional isolated referred material listed by Carroll & Gaskill (1978). I had scored the OTUs this way to test the affinities of the Richards Spur material, but don’t run all three OTUs together. I doubt it will change the topology, but this is a minor detail that needs to be addressed.

Response: These have been removed and replaced simply with a composite *Euryodus dalyae*. The phylogeny has been re-run, and all corresponding sections have been changed.

5. The authors should make sure to double-check the italicization of scientific names in their references because this is something often missed or erroneously altered during

Response: done.

6. I made a few specific comments (e.g., lacrimal-jugal contact, pineal-frontal contact), but there are a few areas where the anatomy as illustrated seems to suggest taphonomic distortion other than just the folding of one side of the skull. It would be helpful for future workers who cannot examine the specimen first-hand to have a more explicit description of the entire physical state of the specimen with this in mind. This is also true of differentiating when something is ‘not exposed’ versus ‘not preserved’ in the text

Line comments: all changes were made.

Reviewer: 2

Comments to the Author(s)

This is an interesting new tetrapod described from the famous Mazon Creek locality in Illinois, and includes a useful discussion of integumentary structures among. I support the eventual publication of this paper but there are a few changes that must take place before that can happen.

The largest issue is the proposed name itself. Article 27 of the ICZN expressly forbids the use of diacritical marks. The proposed name should thus remove the marks above the o, so the name would become Jormungandr or Joermungandr (which is how one represents the modern Swedish phoneme-- I've no idea what the sound would be in Old Norse) or alternatively Jormungand (a common Anglicization). However the authors decide to go I would urge latinizing the name but I don't believe the ICZN mandates this (as long as the language of origin is specified). This change is absolutely mandatory, which is why I gave the paper "major revisions" instead of the "minor revisions" rating it otherwise deserves.

Also mandatory is that the authors must specify gender of the generic name. Modern Swedish does not have male/female genders but Old Norse did as I understand it; regardless this is a proper name that is being proposed, so double check the ICZN to be sure there aren't any specific regulations in this regard and specify your preference explicitly in the Systematic section..

(A more pedantic quibble: the authors also should be cautious when they mention Jörmungandr being the "world serpent" (which I accept is a thing it is called). Jörmungandr is the serpent that dwells in the Midgard sea; Midgard is only one of several worlds in Norse mythology. Nidhoggr is the serpent that feeds on the roots of the World Tree Yggdrasil, which spans all of these other "worlds" (or realms) and is what first jumped to my mind since "world" can be interpreted as "cosmos" in this context. A little more clarification would help avoid others making the same misunderstanding.)

Response: Etymology and name changed to Joermungandr after Swedish. Norse original derivative is noted as gender Masculine. I also changed to Midgard sea serpent! Thanks!

Specific comments:

PDF page 4: citation in abstract

Response: Removed.

page 5: the plural of bauplan is baupläne

Response: changed.

page 9 line 3: these features mentioned setting Odonterpeton aside are all likely ontogenetic.

Response: Perhaps this is true of just the skull shape, and individual postfrontal shape/morphology (although these characters are often still used to distinguish many genera). However, perhaps not so clear-cut is the presence in Odonterpeton, and complete absence in Joermungandr of cheek emargination, where Odonterpeton looks vaguely like that of lysorophians, at more-or less the same size. I don't think the condition seen in Odonterpeton is the same as the simple/small-emargination that happens in some Ostodolepids (Mann et al., 2019, fig 4 for summary).

Similarly, the presence of diminutive, well ossified, limbs in Odonterpeton, as opposed to longer limbs in Joermungandr (humeral length at least) that are clearly not well-developed, while at the same body size seem to indicate difference in limb morphology that is not aligned with uniform ontogenetic pattern in these two animals.

I think the combination of all these features is worth noting here, if only for future studies, so I retained them. I did modify and soften the language in the text to clarify some of these features.

line 34: presumably you mean Quasicaecilia not Cardiocephalus (which has an elongate, bullet-shaped head)?

Response: Yes, changed!

page 10 line 21: the font size changes here (and elsewhere in the MS)

Response: changed, to be uniform.

page 14 line 20: rethink the choice of "worthy", as its not a question of merit but relevance

Response: changed to relevant.

page 17 line 54: replace "revamped" with "recent"

Response: changed.

Figure 5 was too large to download and view in the PDF, oddly since its just a cladogram and can be presented as a greyscale image, so I was not able to evaluate it.

Response: Figure 5 is reduced in size now.

Interesting stuff, I look forward to these mandatory changes being made and consideration of the other points in a future revised MS or the final published paper!

Jason Anderson

Reviewer: 3

Comments to the Author(s)

I support the publication of this manuscript, but feel that quite a lot of additional work is required. Some aspects of the manuscript are in conflict with the illustrations and the available data. For example the authors talk about paired parietals and frontals, but only single elements are shown in the figure. Is the element to the left of the bone identified as frontal actually the left frontal or another element? Possibly, but the skull as preserved is incomplete and it is not clear. The authors refer to a pineal foramen, but this is again not shown in the illustration.

The photographs in Figure 3 are not clear, therefore the interpretations of the cranial anatomy are in doubt. Figure 3B does not show clearly that the element identified as frontal is excluded from the orbit.

I also have trouble with the identification of a short tail. The last caudal is at the tip of the concretion, and there is no evidence that this is the end of the tail.

Lastly, the information on the scales is very interesting and highly informative, but the absence of gastralia is uncertain. I am not familiar with the condition of this kind material, but it appears to be quite 2D, and flattened with only a small part of the outer surface of the fossil being preserved, and when the concretion broke only some of the fossil remains were exposed. In fact, as described and shown, only a small surface of the overall fossil shows the presence of scales. Thus, it is not possible to determine, in my opinion, if gastralia were present or not. I think this is a major weakness in the paper because of the issue of homology of gastralia and the scales, and their relevance to the ecomorphology of the taxon. The description of the scales states that dorsal, lateral, and ventral scales are preserved, but no real evidence of this is provided. It appears as if only the part of the right side of the body fossil shows scales. It is highly unlikely that these impressions of scales represent the dorsal, flank, and ventral regions of the body, and simply stating that without any evidence is not acceptable. I would urge you to reconsider this part of the manuscript and address these concerns.

Response: Figure 3b

Response:

This review is interesting, and gave us a lot to think on, which in turn led to some new observations, so I thank the reviewer in advance for that. Some aspects of the review I could not change because they arise from unfamiliarity with the preservation of the fossils at Mazon Creek, which the reviewer has noted. This is likely why neither of the other reviewers raised these remarks, as both Reviewer 1 & 2 have actually worked on Mazon Creek material before.

Most notably Mazon Creek tetrapod fossils are not preserved in 2D, however, this is a misleading statement by the reviewer to start, because even flattened material, such as tetrapods preserved in cannel-coal blocks at Linton have a 3D quality to their anatomy. Mazon Creek tetrapods are preserved as 3D (unflattened), their osteology is in filled

(secondary diagenesis) with Kaolinite (a clay mineral) and pyrite, these are subsequently removed with preparation to reveal a 3D cast/mold of the anatomy, which can be studied through the use of latex peels and photography/microscope techniques, which was done here. Furthermore, the reviewer should note, because these animals are preserved in 3D, part of the concretion contains the dorsal half, and the other includes the ventral half. That is to say, what we have here is the entire encapsulation of the animals exterior, including all its scales which overlies all the bones. Having studied nearly every known tetrapod concretion from Mazon Creek, I can assure you this is the case, however it's also detailed elsewhere in the literature (e.g. Moodie, 1911; Shabica and Hay, 1995). For further information, I recommend is the reviewer read up on the preservation of fossils from this site (Milner, 1982; Shabica and Hay, 1995; Clements et al., 2019; Mann and Maddin, 2019; Mann et al., 2019; Mann and Gee, 2020 etc. to name a few).

On the topic of preservation, the scales are present encapsulating the entire body—not just the sections indicated on the figure, this is clearly indicated in the text in the description section. This observation is further supported by evidence based on the comparative, and preservational modes present at Mazon Creek. Since the scales line the entire body exterior on dorsal and ventral components (even overlapping the skeletal elements: see figure 4A where the ribs are overlapped by scales!), we can make observation of flank and belly scales which are identified based on their relative position (see other Mazon publications *but also* Hook, 1983). In the figure, where scales are only illustrated on half of the concretion, was stylistically done to not obscure other anatomy, this will now be clearly stated in the figure caption. Interestingly enough re-examination of the specimen prompted by the reviewers' comments has revealed the presence of highly-reduced, gastralia on the ventrum (where they should be if we're encapsulating the entire animal!). These very thin rod-like, gastralia, overlay scales of the ventrum and are now added to the manuscript and thoroughly discussed.

The Illustrations in figure 2 clearly show paired cranial vault elements, and a pineal foramen. The latter will now be labeled. However, I think double labelling the frontals and parietals is almost never done in the paleontological literature—I think this is why neither Reviewer 1 or 2 commented on this. If the reviewer is seriously questioning whether the elements are actually antimeres, I would argue there is no evidence for that assumption, but rather their morphology and position support that the elements are anitmeres. To this effect I also maintain the the postfrontal and prefrontal narrowly exclude the frontal from the orbit, they are just a little shifted in position.

On the short tail. I still think It was probably short due to the rate of caudal shrinkage toward the terminus, and based on that condition in other recumbirostrans. However, because there's a chance we're missing at least some of the tail we have augmented the text, and removed from the character from the diagnosis as suggested.

Appendix C

Review of RSOS-210319 (Mann, Calthorpe & Maddin)

To the authors,

Thank you for resubmitting your revised manuscript; I am pleased to see that the requested changes have been incorporated into the text. I noted a few errors or areas where the wording could be improved below, but in my opinion, the manuscript is now acceptable for publication.

Best wishes,

Bryan Gee
Postdoctoral scholar
University of Washington

Specific comments (pagination refers to Word document, not reviewing proof)

- P. 3, l. 15: ‘preserves’ rather than ‘provides’?
- P. 3, l. 19: “baupläne”?
- P. 3, l. 19-20: consider specifying what is meant by “questioning past ideas about...”; some people may be able to infer from the listed citations that this is about faunal turnover, biogeography, etc., but most may not make that connection
- P. 3, l. 53: there are repeated mentions of FMNH specimens with only FM XXXX
- P. 4, l. 12: ‘one’ versus ‘1’
- P. 5, l. 19: isn’t it multistate scores, not multistate taxa?
- P. 5, l. 38: this is a little confusing because the strict consensus shows the (new) spelling of the taxon as being in a polytomy within specifically Brachystelechidae; if this is the correct position, I would say that, whereas if it is the old spelling, I would say that it is recovered in a polytomy “within Recumbirostra” to avoid implying that Recumbirostra is a single polytomy without resolution
- P. 5, l. 40-47: I am not sure that this large block of text is necessary – the specific relationships of other recumbirostrans are not discussed otherwise, and it seems like a figure reference would suffice
- P. 6, l. 17: should this not include Figure 4?
- P. 6, l. 34: “swedish”
- P. 6, l. 45: extra ‘short’?
- P. 6, l. 50: “remaining distinct” sounds a little awkward – why not just say “distinct, non-coossified supraoccipital” instead?
- P. 7, l. 50: italicization of *Quasicaecilia*
- P. 16, l. 8-9: I would recommend rephrasing this slightly to avoid any confusion that they are actually cranial scales that were covering the head during life (that is not known one way or another) since you make repeated mention of specific positions of scales
- P. 17, l. 48: “upto”

Supplementary Information

- General: spelling of ‘*mckinziei*’
- S1 (majority-rule): both the old and new spellings of *Joermungandr* are listed as OTUs
- S6 (caption): new spelling of *Joermungandr*

Appendix D**ROYAL SOCIETY
OPEN SCIENCE****Joermungandr bolti, an exceptionally preserved 'microsauro'
from Mazon Creek, Illinois, reveals patterns of
integumentary evolution in Recumbirostra.**

Journal:	Royal Society Open Science
Manuscript ID	RSOS-210319
Article Type:	Research
Date Submitted by the Author:	24-Feb-2021
Complete List of Authors:	Mann, Arjan; Carleton University, Earth Sciences Calthorpe, Ami S.; Carleton University, Earth Sciences Maddin, Hillary; Carleton University, Earth Sciences
Subject:	palaeontology < BIOLOGY
Keywords:	Integumentary evolution, Recumbirostra, Carboniferous, Mazon Creek, Scale Ultrastructure, Amniota
Subject Category:	Organismal and Evolutionary Biology

Author-supplied statements

Relevant information will appear here if provided.

Ethics

Does your article include research that required ethical approval or permits?:

This article does not present research with ethical considerations

Statement (if applicable):

CUST_IF_YES_ETHICS :No data available.

Data

It is a condition of publication that data, code and materials supporting your paper are made publicly available. Does your paper present new data?:

Yes

Statement (if applicable):

Additional supporting data are included in the Electronic Supplementary Information. In addition, the nexus file used in the phylogenetic analysis has been deposited in Dryad:

https://datadryad.org/stash/share/4Llz9h6TOqWwuQvRHVa_68t-nf4jM0RtYNcyJ8JPSIY.

Conflict of interest

I/We declare we have no competing interests

Statement (if applicable):

CUST_STATE_CONFLICT :No data available.

Authors' contributions

This paper has multiple authors and our individual contributions were as below

Statement (if applicable):

Arjan Mann wrote the paper, analysed data and constructed figures, Ami Calthorpe aided with figures and data analysis, Hillary Maddin provided edits and comments.

Response to Reviewers Comments:

Editor comments:

Dear Authors: thanks for your submission. As you can see the reviewers are generally positive about the work but each of them has substantial concerns; and, interestingly enough, they are all pretty different from each other. On balance I am giving a "reject/resub" decision mainly because you will have more time to deal with their concerns, not the least of which is the illustrations. Please prepare a response to each concern with your resubmission; we normally send the resubs out for another round of review and so it will be important to meet all the concerns of the reviewers. Best wishes.

Response: Dear editor,

I have addressed the three reviewer's comments to the best of my ability. Virtually every concern of the first and second reviewer have been addressed, and significant changes have been made to the original submission that I hope are to your acceptance. Most of reviewer 3's comments were also addressed and significantly improved the manuscript, however, some of this reviewers' comments and concerns arose from unfamiliarity with the preservation of the material and have not been adopted.

Thanks.

Arjan Mann

See below my detailed responses:

Reviewer comments to Author:

Reviewer: 1

Comments to the Author(s)

To the authors,

I have reviewed the following manuscript describing a new recumbirostran 'microsauro' from Mazon Creek, the latest in a renewed interest in these early tetrapods. The manuscript is well-written and scientifically sound and should be acceptable for publication pending minor revisions. I have attached my detailed comments in the following document (in both PDF and Word formats).

Regards,

Bryan Gee

Response: Thanks for your detailed review, all of the minor revisions have been addressed in your attached Pdf, including most figure changes (save some that were just suggestions) as well as supplemental and coding changes as well. New additions include new phylogenetic analysis and discussion in the description on some tenuous characters. Major/general comments have been addressed, and all line changes have been accepted see below:

General comments

1. There are a number of instances where 'microsauro' is put in single quotation marks, as with most recent workers, but also instances where it is not (e.g., p. 14, l. 16, 28); this should be standardized in

the revision since I doubt it will be during copy-editing.

**Response: changed.**

2. There are numerous instances in which large summary works (particularly Carroll & Gaskill) are
cited as the only reference for a specific statement or figure. Especially for directing readers to
figures that are not reproduced here (the scales of *Hylopleston* is a good example to permit “direct
comparison” by the reader), I would suggest adding the specific figure or pagination, as the authors
did for *Pelodosotis* scales.

**Response: done where applicable.**

3. The author contributions as entered in the submission portal differ slightly from those listed in the
manuscript; the authors should just double check for consistency / their preferred version.

**Response: The manuscript is the preferred version and this will be changed accordingly**

4. As the most recent editor of the utilized matrix, OMNH 53519 is *Euryodus* sp. from Richards
Spur, FMNH UR 2296 is the holotype of *Euryodus dalyae*, and ‘*Euryodus dalyae* (composite)’ is
FMNH UR 2296 + additional isolated referred material listed by Carroll & Gaskill (1978). I had
scored the OTUs this way to test the affinities of the Richards Spur material, but don’t run all three
OTUs together. I doubt it will change the topology, but this is a minor detail that needs to be
addressed.

**Response: These have been removed and replaced simply with a composite *Euryodus dalyae*. The
phylogeny has been re-run, and all corresponding sections have been changed.**

5. The authors should make sure to double-check the italicization of scientific names in their
references because this is something often missed or erroneously altered during

**Response: done.**

6. I made a few specific comments (e.g., lacrimal-jugal contact, pineal-frontal contact), but there are
a few areas where the anatomy as illustrated seems to suggest taphonomic distortion other than just
the folding of one side of the skull. It would be helpful for future workers who cannot examine the
specimen first-hand to have a more explicit description of the entire physical state of the specimen
with this in mind. This is also true of differentiating when something is ‘not exposed’ versus ‘not
preserved’ in the text

**Line comments: all changes were made.**

Reviewer: 2

Comments to the Author(s)

This is an interesting new tetrapod described from the famous Mazon Creek locality in Illinois, and
includes a useful discussion of integumentary structures among. I support the eventual publication of
this paper but there are a few changes that must take place before that can happen.

The largest issue is the proposed name itself. Article 27 of the ICZN expressly forbids the use of
diacritical marks. The proposed name should thus remove the marks above the o, so the name would
become Jormungandr or Joermungandr (which is how one represents the modern Swedish phoneme--
I've no idea what the sound would be in Old Norse) or alternatively Jormungand (a common
Anglicization). However the authors decide to go I would urge latinizing the name but I don't believe
the ICZN mandates this (as long as the language of origin is specified). This change is absolutely
mandatory, which is why I gave the paper "major revisions" instead of the "minor revisions" rating it
otherwise deserves.

Also mandatory is that the authors must specify gender of the generic name. Modern Swedish does not have male/female genders but Old Norse did as I understand it; regardless this is a proper name that is being proposed, so double check the ICZN to be sure there aren't any specific regulations in this regard and specify your preference explicitly in the Systematic section..

(A more pedantic quibble: the authors also should be cautious when they mention Jörmungandr being the "world serpent" (which I accept is a thing it is called). Jörmungandr is the serpent that dwells in the Midgard sea; Midgard is only one of several worlds in Norse mythology. Nidhoggr is the serpent that feeds on the roots of the World Tree Yggdrasil, which spans all of these other "worlds" (or realms) and is what first jumped to my mind since "world" can be interpreted as "cosmos" in this context. A little more clarification would help avoid others making the same misunderstanding.)

Response: Etymology and name changed to Joermungandr after Swedish. Norse original derivative is noted as gender Masculine. I also changed to Midgard sea serpent! Thanks!

Specific comments:

PDF page 4: citation in abstract

Response: Removed.

page 5: the plural of bauplan is baupläne

Response: changed.

page 9 line 3: these features mentioned setting *Odonterpeton* aside are all likely ontogenetic.

Response: Perhaps this is true of just the skull shape, and individual postfrontal shape/morphology (although these characters are often still used to distinguish many genera). However, perhaps not so clear-cut is the presence in *Odonterpeton*, and complete absence in *Joermungandr* of cheek emargination, where *Odonterpeton* looks vaguely like that of lysorophians, at more-or less the same size. I don't think the condition seen in *Odonterpeton* is the same as the simple/small-emargination that happens in some *Ostodolepids* (Mann et al., 2019, fig 4 for summary).

Similarly, the presence of diminutive, well ossified, limbs in *Odonterpeton*, as opposed to longer limbs in *Joermungandr* (humeral length at least) that are clearly not well-developed, while at the same body size seem to indicate difference in limb morphology that is not aligned with uniform ontogenetic pattern in these two animals.

I think the combination of all these features is worth noting here, if only for future studies, so I retained them. I did modify and soften the language in the text to clarify some of these features.

line 34: presumably you mean *Quasicaecilia* not *Cardiocephalus* (which has an elongate, bullet-shaped head)?

Response: Yes, changed!

page 10 line 21: the font size changes here (and elsewhere in the MS)

Response: changed, to be uniform.

page 14 line 20: rethink the choice of "worthy", as its not a question of merit but relevance

Response: changed to relevant.

page 17 line 54: replace "revamped" with "recent"

Response: changed.

Figure 5 was too large to download and view in the PDF, oddly since its just a cladogram and can be presented as a greyscale image, so I was not able to evaluate it.

**Response: Figure 5 is reduced in size now.**

Interesting stuff, I look forward to these mandatory changes being made and consideration of the
other points in a future revised MS or the final published paper!

Jason Anderson

Reviewer: 3

Comments to the Author(s)

I support the publication of this manuscript, but feel that quite a lot of additional work is required.
Some aspects of the manuscript are in conflict with the illustrations and the available data. For
example the authors talk about paired parietals and frontals, but only single elements are shown in
the figure. Is the element to the left of the bone identified as frontal actually the left frontal or another
element? Possibly, but the skull as preserved is incomplete and it is not clear. The authors refer to a
pineal foramen, but this is again not shown in the illustration.

The photographs in Figure 3 are not clear, therefore the interpretations of the cranial anatomy are in
doubt. Figure 3B does not show clearly that the element identified as frontal is excluded from the
orbit.

I also have trouble with the identification of a short tail. The last caudal is at the tip of the concretion,
and there is no evidence that this is the end of the tail.

Lastly, the information on the scales is very interesting and highly informative, but the absence of
gastralia is uncertain. I am not familiar with the condition of this kind material, but it appears to be
quite 2D, and flattened with only a small part of the outer surface of the fossil being preserved, and
when the concretion broke only some of the fossil remains were exposed. In fact, as described and
shown, only a small surface of the overall fossil shows the presence of scales. Thus, it is not possible
to determine, in my opinion, if gastralia were present or not. I think this is a major weakness in the
paper because of the issue of homology of gastralia and the scales, and their relevance to the
ecomorphology of the taxon. The description of the scales states that dorsal, lateral, and ventral
scales are preserved, but no real evidence of this is provided. It appears as if only the part of the right
side of the body fossil shows scales. It is highly unlikely that these impressions of scales represent
the dorsal, flank, and ventral regions of the body, and simply stating that without any evidence is not
acceptable. I would urge you to reconsider this part of the manuscript and address these concerns.

Response: Figure 3b

**Response:**

**This review is interesting, and gave us a lot to think on, which in turn led to some new**
**observations, so I thank the reviewer in advance for that. Some aspects of the review I could**
**not change because they arise from unfamiliarity with the preservation of the fossils at**
**Mazon Creek, which the reviewer has noted. This is likely why neither of the other reviewers**
**raised these remarks, as both Reviewer 1 & 2 have actually worked on Mazon Creek material**
**before.**

**Most notably Mazon Creek tetrapod fossils are not preserved in 2D, however, this is a**
**misleading statement by the reviewer to start, because even flattened material, such as**
**tetrapods preserved in cannel-coal blocks at Linton have a 3D quality to their anatomy.**
**Mazon Creek tetrapods are preserved as 3D (unflattened), their osteology is in filled**

(secondary diagenesis) with Kaolinite (a clay mineral) and pyrite, these are subsequently
removed with preparation to reveal a 3D cast/mold of the anatomy, which can be studied
through the use of latex peels and photography/microscope techniques, which was done
here. Furthermore, the reviewer should note, because these animals are preserved in 3D,
part of the concretion contains the dorsal half, and the other includes the ventral half. That is
to say, what we have here is the entire encapsulation of the animals exterior, including all its
scales which overlies all the bones. Having studied nearly every known tetrapod concretion
from Mazon Creek, I can assure you this is the case, however it's also detailed elsewhere in
the literature (e.g. Moodie, 1911; Shabica and Hay, 1995). For further information, I
recommend is the reviewer read up on the preservation of fossils from this site (Milner, 1982;
Shabica and Hay, 1995; Clements et al., 2019; Mann and Maddin, 2019; Mann et al., 2019;
Mann and Gee, 2020 etc. to name a few).

On the topic of preservation, the scales are present encapsulating the entire body—
not just the sections indicated on the figure, this is clearly indicated in the text in the
description section. This observation is further supported by evidence based on the
comparative, and preservational modes present at Mazon Creek. Since the scales line the
entire body exterior on dorsal and ventral components (even overlapping the skeletal
elements: see figure 4A where the ribs are overlapped by scales!), we can make observation
of flank and belly scales which are identified based on their relative position (see other
Mazon publications *but also* Hook, 1983). In the figure, where scales are only illustrated on
half of the concretion, was stylistically done to not obscure other anatomy, this will now be
clearly stated in the figure caption. Interestingly enough re-examination of the specimen
prompted by the reviewers' comments has revealed the presence of highly-reduced, gastralia
on the ventrum (where they should be if we're encapsulating the entire animal!). These very
thin rod-like, gastralia, overlay scales of the ventrum and are now added to the manuscript
and thoroughly discussed.

The Illustrations in figure 2 clearly show paired cranial vault elements, and a pineal
foramen. The latter will now be labeled. However, I think double labelling the frontals and
parietals is almost never done in the paleontological literature—I think this is why neither
Reviewer 1 or 2 commented on this. If the reviewer is seriously questioning whether the
elements are actually antimeres, I would argue there is no evidence for that assumption, but
rather their morphology and position support that the elements are anitmeres. To this effect
I also maintain the the postfrontal and prefrontal narrowly exclude the frontal from the orbit,
they are just a little shifted in position.

On the short tail. I still think It was probably short due to the rate of caudal shrinkage
toward the terminus, and based on that condition in other recumbirostrans. However,
because there's a chance we're missing at least some of the tail we have augmented the text,
and removed from the character from the diagnosis as suggested.

TITLE PAGE

Jörmungandr bolti, an exceptionally preserved ‘microsaur’ from the Mazon Creek Lagerstätte reveals patterns of integumentary evolution in Recumbirostra.

Formatted for Royal Society Open Science

Arjan Mann^{1*}, Ami S. Calthorpe¹, and Hillary C. Maddin¹

¹Department of Earth Sciences, Carleton University, 2115 Herzberg Laboratories, 1125 Colonel By Drive, Ottawa, Ontario, K1S 5B6, Canada, arjan.mann@carleton.ca*

Running title: Exceptionally preserved ~~The exceptionally preserved, Joe~~*örmungandr bolti*

*Corresponding author: Arjan Mann; Department of Earth Sciences, Carleton University, 1125 Colonel By Drive, Ottawa, Ontario K1S 5B6, Canada; email: arjan.mann@carleton.ca

ABSTRACT

The Carboniferous Pennsylvanian-aged (309-307 Ma) Mazon Creek Lagerstätte produces some of the earliest fossils of major Paleozoic tetrapod lineages. Recently, several new tetrapod specimens collected from Mazon Creek have come to light, including the earliest fossorially-adapted recumbirostrans. Here we describe a new long-bodied recumbirostran, *Joeërmungandr bolti* gen et sp. nov., known from a single part and counterpart concretion bearing a virtually complete skeleton. Uniquely, *Joeërmungandr* preserves a full suite of dorsal, flank, and ventral dermal scales, together with a series of thinned and reduced ~~but lacks any trace of~~ gastralia. Investigation of these scales using Scanning Electron Microscopy (SEM) reveals ultrastructural ridge and pit morphologies, revealing complexities comparable to the scale ultrastructure of extant snakes and fossorial reptiles, which have scales modified for body based propulsion and shedding substrate. Our new taxon also represents an important early record of an elongate recumbirostran bauplan, wherein several features linked to fossoriality, including a characteristic recumbent snout, are present. We utilized parsimony phylogenetic methods to conduct phylogenetic analyses using the most recent updated ~~recumbirostran-focused matrix of Gee et al., (2020)~~. The analysis recovers s-*Joermungandr* within Recumbirostra with likely affinities to the new taxon within the Cocytinoidea clade ~~as sister taxon to all brachystelechids~~. Finally, we review integumentary patterns in Recumbirostra, noting reductions and losses of gastralia and osteoderms associated with body elongation and, thus, likely also associated with increased fossoriality.

INTRODUCTION

Recent resurgence in the study of certain ‘microsaurs’ known as recumbirostrans has resulted in their renewed relevance in the origin of amniotes (Vaughn, 1964; Pardo et al., 2017). Debates specifically concern whether Recumbirostra belong to the amniote stem group (Laurin and Reisz, 1995; Reisz et al., 2015), or are instead crown amniotes derived from a reptilian ancestor (Pardo et al., 2017). The timing of the origin of amniotes is earmarked at the early mid-Pennsylvanian ~~Carboniferous~~, with the oldest unambiguous amniotes residing in the Bashkirian-aged (~3185 Ma) strata of Joggins, Nova Scotia (Mann et al., 2020). Already at this stage, and onward through the Permo-Carboniferous, recumbirostrans are morphologically diverse,

showing a range in degree of development of a variety of stereotypical cranial and postcranial
adaptations for a fossorial lifestyle (Huttenlocker et al., 2013; Pardo et al., 2015; Szostakiwskyj
et al., 2015; Pardo and Anderson, 2016; Mann et al., 2017). Thus the inclusion of Recumbirostra
into Eureptilia would redefine the nature of the radiation of the groups and reveal details of the
ecology and diversity in some of the earliest phases of reptile evolution. Recent studies (Mann
and Maddin, 2019; Mann et al., 2019a; Mann et al., 2019b; Mann and Gee, 2020) on tetrapod
fossils from the Moscovian-aged (309-307 Ma) Mazon Creek Lagerstätte have revealed a diverse
'recumbirostran-amniote' assemblage fauna. Because Mazon Creek often provides fossils of
entire organisms, including soft-tissue structures entombed within siderite concretions, a diverse
array of bauplänebauplans have been preserved, questioning past ideas about terrestrial tetrapod
diversity in Carboniferous ecosystems (Milner, 1982; Ahlberg and Milner, 1984; Sues and Reisz,
1998; Sahney et al., 2010; Dunne et al., 2018; but see Pardo et al., 2019).

Here we describe *Joeërmungandr bolti* gen. et sp. nov., a new long-bodied
recumbirostran from Mazon Creek, Francis Creek Shale, Illinois (Fig. 1–3). *Joeërmungandr*
provides an ecomorphological intermediate between the extreme body elongation and limb
reduction seen in molgophid recumbirostrans such as the contemporaneous *Infernovenator* and
the short-bodied, robustly built brachystelchid recumbirostran *Diabloroter* (Mann and Maddin,
2019; Mann et al., 2019a). *Joeërmungandr* is exceptionally preserved, at the higher end of the
quality scale for Mazon Creek fossils. It includes alongside the full skeleton visible in dorsal and
ventral aspects, a detailed soft tissue impression of the body complete with scales. Detailed
examination of the scales of *Joeërmungandr* reveal a unique ultrastructural pattern, and provide
the first comprehensive description of scale morphology of this kind in a Paleozoic tetrapod (Fig.
4).

MATERIALS AND METHODS

Specimens were studied at: Augustana College's Fryxell Geology Museum (ACFGM),
Rock Island, USA; American Museum of Natural History (AMNH), New York, USA; Carnegie
Museum of Natural History (CM), Pittsburgh, USA; Denver Museum of Nature and Science
(DMNH), Denver, USA; Field Museum of Natural History (FMNH), Chicago, USA; University
of Kansas Natural History Museum (KUVN), Lawrence, USA; University of Nebraska State

Museum (UNSM), Lincoln, USA; Smithsonian Institution (USNM), Washington DC, USA; and Yale Peabody Museum (YPM), New Haven, USA. Additional comparative specimens from the British Museum of Natural History (BMNH), London, UK, the Harvard Museum of Comparative Zoology (MCZ), Cambridge, USA, Museum für Naturkunde (MB), Berlin, Germany, were compared based on casts, latex peels, and existing literature.

Only 1 latex peel was made of the dorsal aspect of the skeleton, and is catalogued with the original material at the FMNH. This latex cast was used to describe the dorsal cranial and postcranial elements; however, the ventral half of the skeleton is described solely off of the corresponding counterpart. Photography was conducted using a Sony Alpha ILCE 5000 camera, F3.5 lens. All figures were drawn and formatted in Photoshop CS6 (Adobe, San Jose, CA). For phylogenetic methodology see the phylogenetic analysis section below.

In order to interpret features of the morphology of the scales preserved on *Joeörmungandr bolti*, a extant comparative dataset was examinedeompiled, consisting of extant squamate scales belonging to a number of terrestrial, arboreal and fossorial forms housed in the research collections of the Canadian Museum of Nature, Gatineau, Quebec (CMN). A list of The selected extant squamate specimens can be found in supplementary data. Scanning Electron Microscopy (SEM) was conducted at the Canadian Museum of Nature research facility using a JEOL 6610LV SEM.

Anatomical abbreviations

ang=angular; **bo**=basioccipital; **co**=co-ossified occipital and otic elements; **dv**=dorsal vertebrae; **dr**=dorsal ribs; **d**=dentary; **ec**=ectopterygoid; **f**=frontal; **fe**=femur; **fib**=fibula; **gb**=gastric bolus; **gs**=gastralia; **h**=humerus; **il**=ilium; **j**=jugal; **l**=lacrima; **m**=maxilla; **mt**=metatarsal; **n**=nasal; **p**=parietal; **pas**=parasphenoid; **pf**=pineal foramen; **pof**=postfrontal; **pmx**=premaxilla; **prf**=prefrontal; **ptl**=pterygoid alatine; **qj**=quadratojugal; **so**=supraoccipital; **sq**=squamosal; **sv**=sacral vertebra; **sp**=splenial; **t**=tabular; **tib**=tibia; **u**=ulna; **v**=vertebrae.

PHYLOGENETIC ANALYSIS

We explored the phylogenetic relationships of *Joermungandr bolti* FMNH 1309 using a modified version of the recent matrix of Pardo *et al.* (2017; see Supplementary Information file for matrix), which provides the most up to date matrix for assessing recumbirostran interrelationships. The matrix used has been taxonomically modified in recent studies by Mann and Maddin (2019), Mann *et al.*, (2019), and Gee *et al.*, (2020). These modifications were retained in the current analysis. We performed a parsimony analysis using PAUP software v4.0b10 (Swofford, 2002) and *Eusthenopteron* was specified as the outgroup. We used the heuristic search option with the TBR search algorithm and 1000 random addition sequence replicates. Maxtrees was set at 10,000, and automatically increased by 100. All characters were equally weighted. All multistate taxa were treated as polymorphic. All ambiguous character states were resolved using the ACCTRAN setting. Indices of goodness of fit of the character data to the topology (e.g., consistency index [CI], retention index [RI], rescaled consistency index [RC], and homoplasy index [HI]) were calculated in PAUP. To assess support of internal nodes, bootstrap values were calculated using the fast stepwise addition option with 1000 replicates.

The parsimony analysis recovered 270-18 most parsimonious trees (MPT), each with 185053 steps (CI = 0.3065 ; HI = 0.7488; RI = 0.64957; RC = 0.199204) (see Supplemental Information for all phylogenetic analyses; S1–3). The majority rule consensus of the results recovered *Joermungandr* as sister taxa to the Cocytinoidea, a clade that includes the sister taxon relationship between molgophids and brachystelechids. The majority rule consensus also recovers most previously reported relationships (e.g. Pardo *et al.*, 2017). The strict consensus of the results recovered *Joermungandr* in a polytomy with all recumbirostrans crownward of *Pantylus*. These include: *Cardiocephalus peabodyi*; *Cardiocephalus sternbergi*; *Pariotichus brachyops*; *Euryodus dalyae*; *Euryodus primus*; *Proxilodon bonneri*; *Huskerpeton englehorni*; *Rhynchonkos stovalli*; a clade containing *Llistrofus* to the exclusion of ostodolepids; a clade where molgophids are sister taxa to brachystelechids; and finally a clade containing the sister taxa *Dvellecanus carrolli* and *Aletrimyti gaskillae*. FMNH 1309 as sister taxon to the clade including all brachystelechids (Fig. 5). This relationship is supported by five characters: 17(0), 28(2), 47(2), 48(0), 53(1), 115(0), 139(1), and 267(0)~~13-state=1, 29-state=0, 52-state=1, 75-state=2, and 267-state=0~~ (all character numbers refer to those used in the analysis of Pardo *et al.*, 2017). The All-~~b~~bootstrap tree and associated values can also be found in the Supplemental Information (S3) above 50% are reported on Figure 5.

Systematic Palaeontology

Tetrapoda Jaekel, 1909

Recumbirostra Anderson, 2007

Joeörmungandr bolti gen. et sp. nov.

(Figs. 1–3)

Zoobank LSID: Will be provided with publication.

Holotype. FMNH 1309, part and counterpart of a siderite concretion containing a virtually complete skeleton and soft body impression in dorsal and ventral views.

Locality and Horizon: Mazon Creek, Grundy County, Illinois, USA. Francis Creek Shale, above the Morris (no. 2) Coal, Carbondale Formation, Middle Pennsylvanian (Moscovian).

Etymology. ‘*Joeörmungandr*’ is the ~~name given to the~~ swedish phoneme of ‘Jörmungandr’ (gender: masculine) the name of the serpent that dwells in the ‘midgard sea’ world serpent from Norse mythology. The specific epithet ‘*bolti*’ is in honour of the late paleontologist John R. Bolt.

Diagnosis. A small recumbirostran with the following unique combination of characters: 40 presacral vertebrae; ~~short tail with 17 caudal vertebrae;~~ dentary and maxillary tooth rows terminate anteriorly, occupying approximately half the length of each respective element; -short; short snout; large rectangular parietal that reaches the postorbital and squamosal; large quadrangular postfrontal that invades the anterior margin of the parietal; co-ossification of otic capsules and occiput with —supraoccipital remaining distinct; body covered in uniform scales; scales ornamented with linear ridges and unevenly distributed pits.

[revised manuscript text omitted]

Careful examination of the scales of both recumbirostrans and other ‘microsaurs’ reveals
this is certainly not the case – the scales on *Joeërmungandr* are just one example to the contrary.
Ornamented dermal scales are also well-represented on specimens of *Microbrachis*, *Hyloplecion*
(at present including FMNH PR 981), *Sparodus*, *Crinodon*, *Pelodosotis*, *Llistrofus* (see Gee *et*
*al.*, 2019), among others. Comparatively, only a few temnospondyl groups have convergently
evolved similarly ornate scales to those of ‘microsaurs’. Anastomosing longitudinal striae can be
observed on micromelerpetontids and some amphibamiformes (Mann and Gee, 2020; Witzmann,
2007). Among fossorially-adapted recumbirostrans, such as *Joeërmungandr*, it is possible that
the distinct ridge patterns and other ultrastructural details are adaptations to burrowing where
they would help shed substrate, similar to what is seen in extant fossorial reptile scales (Gans and
Baic, 1977).

Carroll and Gaskill (1978) noted that there are likely systematic differences in scale
morphology across the many microsaurian families. However, at the time the infrequent
occurrence of scales allowed only for morphological descriptions and not for assessment of
evolutionary patterns. What morphological patterns of scalature that can now be summarized for
Recumbirostra are discussed here and reveal some clade level distinctions (Fig. 56). Early
diverging, more generalised recumbirostrans such as *Steenerpeton* (formerly *Asaphestera*
*intermedia*, see Mann et al., 2020) bear large, smooth, plate-like ovular scales that sometimes
show concentric rings (possibly growth rings). This morphology is rare aberrant among
recumbirostrans and may represent the plesiomorphic condition. Hapsidoparideontidids, which
appear to occupy a basal position in recumbirostran phylogeny (Mann et al., 2019b; Gee *et al.*,
201920), have scales known from *Hapsidoparieon*, *Llistrofus*, and *Saxonerpeton*. The ovular
scales of *Llistrofus* (see Gee *et al.*, 2019) are highly ornamented bearing ridges that form a
radiating, clam shell-like pattern, sometimes with a bony webbing connecting ridges. The scales
of *Llistrofus* can also bear a thicker raised margin on either of the anterior or posterior
articulating ends. The scales of *Crinodon* appear morphologically similar to *Llistrofus*, however,
placement of this 'microsauro' is currently uncertain. Ostodolepids show interesting variations in
scale morphology with very fine, 'matted' scales present on *Pelodostis* (see Fig. 130C from
Carroll and Gaskill [1978]), whereas *Micraroter* (BPI 3839) bears very thick dermal scales that
are even thicker on the ventral parts of the animal. Ornamentation may be present on these
scales, however, they are not preserved well enough to parse out any details of the pattern.

Among Joermungandr and the the Cocytinoidea (Brachystelechidae + Molgophidae),
there appears brachystelechids and Jörmungandr appear to bear similar types of dermal scales
that are very thin with ridged ornamentation. Molgophids appear to also bear thin dermal scales,
but the details of these are not well-preserved on any known specimen (Mann *et al.*, 2019). Of
the Pantylidae, *Sparodus* has the best examples of an ornate scale morphology consisting of fine
ridges and perpendicular, evenly spaced, concentric circles (Carroll and Gaskill, 1978). No
specimen of *Pantylus* shows dermal scalation, save one specimen (FM UR 1069) that only bears
fragments. However, given this specimen and that the other members of this clade possess
scales, it is likely *Pantylus* did as well. *Trachystegos* is heavily scaled, although its referral to
Pantylidae is at present questionable. Scales are not known in any member of the Rhyntonkidae,
or curiously any member of the Gymnarthridae (Carroll and Gaskill, 1978). Although scales are

often found preserved alongside gymnarthrids at Joggins, their association with a gymnarthrid is not certain due to the fragmentary nature of ‘microsaur’ material at the site and the presence of multiple tetrapods within a single stump. Recently, scales were reported by Gee *et al.* (2020) on the skull roof of the gymnarthrid *Euryodus*.

Furthermore, neither fossils of *Cardiocephalus* nor fossils of *Euryodus* preserve any remains of scales. Because ornamented scales phylogenetically bracket this group, it is likely that they were present in these animals as well and that the conditions where these fossils are found did not favour the preservation of such delicate structures.

The morphology of both dorsal and ventral scales are identical to one another in *Joermungandr*. Some recumbirostrans, however, show considerable variation in dermal scalation across the body, typically between the dorsal and ventral scales. For example, in *Micraroter* the ventral scales are considerably thickened in comparison to the dorsal scales (Carroll and Gaskill, 1978). Ventral scales differentiation in the aforementioned recumbirostrans are still dermal scale in origin and not homologous with gastralia. That being said, there are a few cases where amniote-like gastralia are also found in recumbirostrans. The early recumbirostran *Steenerpeton* appears to bear a scattering of cylindrical gastralia (Mann *et al.*, 2020). Similarly, gastralia are also present in articulation on *Joermungandr* and the short-bodied brachystelechids, e.g. *Diabloroter* and *Batropetes*, where they appear as thin, cylindrical structures, tapering on either side, that that are morphologically similar to the gastralia of basal amniotes (Mann and Maddin, 2019). However, the gastralia of these recumbirostrans are markedly reduced, forming sparse, string-like, rows of ossifications that occupy a smaller surface area on the ventrum. This contrasts the chevrons of gastralia present in early amniotes such as *Cephalerpeton* (Carroll and Baird, 1972; Reisz, 1975; Mann *et al.*, 2019b), that form a continuous ventral matting of bone.

Given recent ~~vamped~~ phylogenetic hypotheses that recumbirostrans are in fact amniotes descended from a captorhinid-like ancestor (Pardo *et al.*, 2017), it is possible that recumbirostrans at some point had true gastralia (Figs. 5-6). In this scenario basal recumbirostrans such as *Steenerpeton* would retain the plesiomorphic condition for Recumbirostra, whereas *Joermungandr* and brachystelechids reveal a reversal possibly associated with a unique ecology (Fig. 56; Mann and Maddin, 2019). This starkly contrasts the hypothesis of Carroll and Gaskill (1978) who considered the order Microsauria to be stem-

amniotes (Lepospondyli). In this scenario, their morphology was considered as derived from a
basal tetrapod antecedent.

In addition to dermal scales and gastralia, Carroll and Gaskill (1978) noted the presence
of another kind of integumentary structure, bony ‘ossicles’ (a term used for small osteoderms),
along the ventral regions of certain shorter bodied ‘microsaurs’ (Fig. 56). *Pantylus*, *Stegotretus*,
and *Saxonerpeton* all bear unique hexagon-shaped osteoderms that line the ventral pectoral
region of the animal (Fig. 56). Personal examination of these ossicles on *Pantylus*, *Stegotretus*,
and an unnamed pantylid from Nova Scotia, reveal a dense assembly of these osteoderms that
form a mosaic within the inter-dentary space extending posterior to the pectoral girdle (when
preserved). These ossicles bear resemblance to the inter-dentary osteoderms found on *Tuditanus*,
*Crinodon* and *Cardiocephalus* and are likely homologous structures (Carroll and Gaskill, 1978).
Carroll (1968) proposed these osteoderms were derived from the ventral gastralia, however, we
note that these ventral scales are not morphologically (small ossicles in a mosaic) or positionally
(pectoral to inter-dentary space) similar to ventral gastralia, and are likely instead independently
derived ossifications.

Unlike other Permo-Carboniferous groups such as dissorophids and chroniosuchians
(Witzmann, 2007), recumbirostrans do not possess ‘bulky’ or large osteoderms. This is likely
due to the functional constraints associated with a fossorial lifestyle, where large osteoderms
protruding from the dermis would hinder both the locomotion and flexibility needed to achieve
burrowing. Instead the bodies of recumbirostrans such as *Joeörmungandr* appear from body
outline impressions to have been streamlined, cylindrical, and relatively smooth. Extant fossorial
reptiles also have streamlined bodies that appear relatively smooth (e.g. amphisbaenians),
lacking large osteoderms or other protruding keratinous scales or structures (Gans and Baic,
1977).

Finally, body-elongation associated with increased fossoriality is common among
recumbirostrans (only absent in brachystelechids, pantylids, and basal recumbirostrans) and even
reaches extreme lengths in molgophids such as *Brachydectes* (upto 99 presacral vertebrae).
However, similar to the lack of osteoderms or ‘bony-ossicles’ in longer bodied recumbirostrans
such as *Joeörmungandr* and molgophids, there is also an reduction and in some cases complete
absence of gastralia (Fig. 56). Long-bodied recumbirostrans such as *Joeörmungandr* likely relied
on lateral undulation or maybe some form of sidewinding (Cotena and Hembree, 2014) as a

locomotory mode, and a reduction of ventral ossifications including ~~loss of~~ the gastralia may
have aided in providing flexibility to the ventrum (Fig. 56).

**Ethics**

No ethics assessment was required prior to the completion of this research because this study
relied entirely on museum collections. Similarly, collecting permits were not required because no
field collections were made.

**Data accessibility**

Additional supporting data are included in the Electronic Supplementary Information. In
addition, the nexus file used in the phylogenetic analysis has been deposited in Dryad:
https://datadryad.org/stash/share/4Llz9h6TOqWwuQvRHVa_68t-nf4jM0RtYNcyJ8JPSIY.

**Authors' contributions**

30 A.M. designed the study, A.M. and A.S.C. collected and analysed the data, A.M. and H.C.M.
wrote the paper.

**Competing interests**

All authors declare no competing interests.

**Funding**

This project was partially funded by an Ontario Graduate Scholarship grant awarded to Arjan
Mann.

**ACKNOWLEDGEMENTS**

We would like to thank William Simpson, Adrienne Stroup, Diane Scott, Robert Reisz,
Dave Berman, Amy Henrici, and Robert Hook for access to comparative material and providing

helpful discussion. We thank ~~Bryan Gee~~, Jason Pardo, Emily McDaniel for providing discussion,
support, and help with figure construction. ~~Finally, We~~ thank and are indebted to the late John
~~R.~~ Bolt for his kindness and generosity with sharing his knowledge and material. Finally, we
thank Jason Anderson, Bryan Gee, and another anonymous reviewer for helpful comments on
this manuscript.

REFERENCES

- 1. Abdel-Aal, H. A. (2018). Surface structure and tribology of legless squamate reptiles.
*Journal of the mechanical behavior of biomedical materials*, 79, 354-398.
- 2. Anderson, J. S. (2007). Incorporating ontogeny into the matrix: a phylogenetic evaluation
of developmental evidence for the origin of modern amphibians; pp. 182–227 in J. S.
Anderson and H.-D. Sues (eds.), *Major Transitions in Vertebrate Evolution*. Indiana
University Press, Bloomington, Indiana.
- 3. Ahlberg, P. E., & Milner, A. R. (1994). The origin and early diversification of tetrapods.
*Nature*, 368(6471), 507-514.
- 4. Baur -G 1889. *Palaeohatteria* Credner, and the Proganosauria. *American Journal of*
*Science* 37: 310–313.
- 5. Bedford, G. S., & Christian, K. A. (1996). Tail morphology related to habitat of varanid
lizards and some other reptiles. *Amphibia-Reptilia*, 17(2), 131-140.
- 6. Buchwitz, M., Witzmann, F., Voigt, S., & Golubev, V. (2012). Osteoderm microstructure
indicates the presence of a crocodylian-like trunk bracing system in a group of armoured
basal tetrapods. *Acta Zoologica*, 93(3), 260-280.
- 7. Claessens LPAM 2004. Dinosaur gastralgia; origin, morphology, and function. *Journal of*
*Vertebrate Paleontology* 24: 89–106.
- 8. Catena, A., **Hembree, D.I.**, 2014. Swimming through the substrate: the neoichnology of
*Chalcides ocellatus* and biogenic structures of sand-swimming vertebrates.
*Palaeontologia Electronica*, v. 17.3.37A, p. 1-19.
- 9. Castanet J, Francillon-Vieillot H, De Ricqlès A, Zylberberg L 2003. The skeletal
histology of the Amphibia. In: Heatwole H, Davies M, eds *Amphibian Biology*, Vol. 5
*Osteology*. Chipping Norton: Surrey Beatty & Sons, 1598–1683.

10. Daniels, C. B. (1984). The importance of caudal lipid in the gecko *Phyllodactylus marmoratus*. *Herpetologica*, 337-344.
11. Dunne, E. M., Close, R. A., Button, D. J., Brocklehurst, N., Cashmore, D. D., Lloyd, G. T., & Butler, R. J. (2018). Diversity change during the rise of tetrapods and the impact of the ‘Carboniferous rainforest collapse’. *Proceedings of the Royal Society B: Biological Sciences*, 285(1872), 20172730.
12. Gans, C., & Baic, D. (1977). Regional specialization of reptilian scale surfaces: relation of texture and biologic role. *Science*, 195(4284), 1348-1350.
13. Gee, B. M., Bevitt, J. J., & Reisz, R. R. (2020). Computed tomographic analysis of the cranium of the early Permian recumbirostran ‘microsaur’ *Euryodus dalyae* reveals new details of the braincase and mandible. *Papers in Palaeontology*.
14. Gee, B. M., Bevitt, J. J., Garbe, U., & Reisz, R. R. (2019). New material of the ‘microsaur’ *Llistrofus* from the cave deposits of Richards Spur, Oklahoma and the paleoecology of the Hapsidopareiidae. *PeerJ*, 7, e6327.
15. Hirasawa, T., & Kuratani, S. (2015). Evolution of the vertebrate skeleton: morphology, embryology, and development. *Zoological letters*, 1(1), 2.
16. Howes GB, Swinnerton HH 1901. On the development of the skeleton of the Tuatara, *Sphenodon punctatus* with remarks on the egg, on the hatching and on the hatched young. *Transactions of the Zoological Society of London* 16: 1–86.
17. Hook, R. W. (1983). *Colosteus scutellatus* (Newberry): a primitive temnospondyl amphibian from the Middle Pennsylvanian of Linton, Ohio. *American Museum novitates*; no. 2770.
18. Huttenlocker, A. K., Pardo, J. D., Small, B. J., & Anderson, J. S. (2013). Cranial morphology of recumbirostrans (Lepospondyli) from the Permian of Kansas and Nebraska, and early morphological evolution inferred by micro-computed tomography. *Journal of Vertebrate Paleontology*, 33(3), 540-552.
19. Jackson, M. K., & Reno, H. W. (1975). Comparative skin structure of some fossorial and subfossorial leptotyphlopoid and colubrid snakes. *Herpetologica*, 350-359.
20. Jaekel, O. (1909). Über die Klassen der Tetrapoden [About the classes of the tetrapods]. *Zoologischer Anzeiger* 34:193–212.

21. Klein, M. C. G., & Gorb, S. N. (2014). Ultrastructure and wear patterns of the ventral epidermis of four snake species (Squamata, Serpentes). *Zoology*, *117*(5), 295-314.
22. Laurin, M., & Reisz, R. R. (1995). A reevaluation of early amniote phylogeny. *Zoological Journal of the Linnean Society*, *113*(2), 165-223.
23. Mann, A. (2018). Cranial ornamentation of a large *Brachydectes newberryi* (Recumbirostra: Lysorophia) from Linton, Ohio. *Vertebrate Anatomy Morphology Palaeontology*, *6*, 91-96.
24. Mann, A., Pardo, J. D., & Maddin, H. C. (2019a). *Infernovenator steenae*, a new serpentine recumbirostran from the 'Mazon Creek' Lagerstätte further clarifies lysorophian origins. *Zoological Journal of the Linnean Society*, *187*(2), 506-517.
25. Mann, A., & Maddin, H. C. (2019). *Diabloroter bolti*, a short-bodied recumbirostran 'microsaur' from the Francis Creek Shale, Mazon Creek, Illinois. *Zoological Journal of the Linnean Society*, *187*(2), 494-505.
26. Mann, A., McDaniel, E. J., McColville, E. R., & Maddin, H. C. (2019b). *Carbonodraco lundi* gen et sp. nov., the oldest parareptile, from Linton, Ohio, and new insights into the early radiation of reptiles. *Royal Society Open Science*, *6*(11), 191191.
27. Mann, A., & Gee, B. M. (2020). Lissamphibian-like toepads in an exceptionally preserved amphibamiform from Mazon Creek. *Journal of Vertebrate Paleontology*, e1727490.
28. Milner, A. R. (1987). The Westphalian tetrapod fauna; some aspects of its geography and ecology. *Journal of the Geological Society*, *144*(3), 495-506.
29. Miralles, A., Köhler, J., Vieites, D. R., Glaw, F., & Vences, M. (2011). Hypotheses on rostral shield evolution in fossorial lizards derived from the phylogenetic position of a new species of *Paracontias* (Squamata, Scincidae). *Organisms Diversity & Evolution*, *11*(2), 135-150.
30. Moodie, R. L. (1909). *Carboniferous air-breathing vertebrates of the United States National Museum* (Vol. 37).
31. Pardo, J. D., Szostakiwskyj, M., Ahlberg, P. E., & Anderson, J. S. (2017). Hidden morphological diversity among early tetrapods. *Nature*, *546*(7660), 642-645.
32. Pardo, J. D., Szostakiwskyj, M., & Anderson, J. S. (2015). Cranial morphology of the brachystelechid 'microsaur' *Quasicaecilia texana* Carroll provides new insights into the

diversity and evolution of braincase morphology in recumbirostran ‘microsaurs’. *PloS*
*one*, 10(6).
33. Pardo, J. D., & Anderson, J. S. (2016). Cranial morphology of the Carboniferous-Permian
tetrapod *Brachydectes newberryi* (Lepospondyli, Lysorophia): new data from μ CT. *PloS*
*one*, 11(8).
34. Pardo, J. D., Small, B. J., Milner, A. R., & Huttenlocker, A. K. (2019). Carboniferous–
Permian climate change constrained early land vertebrate radiations. *Nature ecology &*
*evolution*, 3(2), 200-206.
35. Reisz, R. R., LeBlanc, A. R., Sidor, C. A., Scott, D., & May, W. (2015). A new
captorhinid reptile from the Lower Permian of Oklahoma showing remarkable dental and
mandibular convergence with microsaurian tetrapods. *The Science of Nature*, 102(9-10),
50.
36. Romer AS 1956. *Osteology of the Reptiles*. Chicago: University of Chicago Press.
37. Ruibal, R. (1968). The ultrastructure of the surface of lizard scales. *Copeia*, 698-703.
38. Sahney, S., Benton, M. J., & Falcon-Lang, H. J. (2010). Rainforest collapse triggered
Carboniferous tetrapod diversification in Euramerica. *Geology*, 38(12), 1079-1082.
39. Szostakiwskyj, M., Pardo, J. D., & Anderson, J. S. (2015). Micro-CT study of
*Rhynchonkos stovalli* (Lepospondyli, Recumbirostra), with description of two new
genera. *PLoS One*, 10(6).
40. Sues, H. D., & Reisz, R. R. (1998). Origins and early evolution of herbivory in tetrapods.
*Trends in Ecology & Evolution*, 13(4), 141-145.
41. Taylor, T. N., & Scott, A. C. (1983). Interactions of plants and animals during the
Carboniferous. *Bioscience*, 33(8), 488-493.
42. Vaughn, P. P. (1962). The Paleozoic microsaurs as close relatives of reptiles, again.
*American Midland Naturalist*, 79-84.
43. Voeltzkow A, Döderlein L 1901. Beiträge zur Entwicklungsgeschichte der Reptilien III.
Zur Frage nach der Bildung der Bauchrippen. *Abhandlungen der Senckenbergischen*
*Naturforschenden Gesellschaft* 26: 313–336.
44. Witzmann, F. (2011). Morphological and histological changes of dermal scales during the
fish-to-tetrapod transition. *Acta Zoologica*, 92(3), 281-302.

45. Witzmann, F. (2007). The evolution of the scalation pattern in temnospondyl amphibians.
*Zoological Journal of the Linnean Society*, 150(4), 815-834.

**Figure captions**

**Figure 1.** Photographs of the holotype of *Jörmungandr bolti* gen. et sp. nov. (FMNH 1309). A,
the part specimen showing the dorsal view; B, the counterpart specimen showing the ventral
view.

**Figure 2.** Illustrations of the holotype of *Jörmungandr bolti* gen. et sp. nov. (FMNH 1309) A,
dorsal and B, ventral aspects of the holotype of *Jörmungandr bolti* gen. et sp. nov. (FMNH
1309).

**Figure 3:** Illustrations of the A-B, dorsal and and C-D, ventral aspects of the skull of
*Jörmungandr bolti* (FMNH 1309). Illustrations have been digitally ~~mirrored reversed~~ in
~~P~~photoshop to show true anatomical orientations.

**Figure 4:** A-C, Scanning Electron Micrographs (SEMs) of *Jörmungandr bolti* (FMNH 1309),
showing progressively closer structures of the dermal scales. D, an idealised illustration of a
single dermal scale (not to scale).

**Figure 5:** ~~Results of the parsimony phylogenetic analysis showing the position of *Jörmungandr*~~
~~*bolti* (FMNH 1309) as sister taxon to all brachystelechids. Bootstrap values are located on top of~~
~~nodes (only those over 50 reported). Total group Recumbirostra indicated in orange and total-~~
~~group Amniota is indicated in red.~~

**Figure 6:** Simplified phylogeny of recumbirostrans showing integumentary structures and body
elongation. Body elongation is indicated in silhouettes with grey=no body elongation;

black=body elongation present. Question marks in boxes indicate uncertain character states
rather than absences. Silhouettes for Ostodolepidae, Gymnarthridae, Rhynchonkidae and
Synapsida modified from Pardo *et al.* (2017).

Figure 1. Photographs of the holotype of *Jörmungandr bolti* gen. et sp. nov. (FMNH 1309). A, the part specimen showing the dorsal view; B, the counterpart specimen showing the ventral view.

165x125mm (600 x 600 DPI)

Figure 2. Illustrations of the holotype of *Jörmungandr bolti* gen. et sp. nov. (FMNH 1309) A, dorsal and B, ventral aspects of the holotype of *Jörmungandr bolti* gen. et sp. nov. (FMNH 1309).

Figure 3: Illustrations of the A-B, dorsal and and C-D, ventral aspects of the skull of *Jörmungandr bolti* (FMNH 1309). Illustrations have been digitally mirrored in Photoshop to show true anatomical orientation.

Figure 4: A-C, Scanning Electron Micrographs (SEMs) of *Joermungandr bolti* (FMNH 1309), showing progressively closer structures of the dermal scales. D, an idealised illustration of a single dermal scale (not to scale).

Figure 5: Simplified phylogeny of recumbirostrans showing integumentary structures and body elongation. Body elongation is indicated in silhouettes with grey=no body elongation; black=body elongation present. Question marks in boxes indicate uncertain character states rather than absences. Silhouettes for Ostodolepidae, Gymnarthridae, Rhynchonkidae and Synapsida modified from Pardo et al. (2017).

Appendix E

Response to reviewers

Reviewer comments to Author:

Reviewer: 1

Comments to the Author(s)

Thank you for resubmitting your revised manuscript; I am pleased to see that the requested changes have been incorporated into the text. I noted a few errors or areas where the wording could be improved below, but in my opinion, the manuscript is now acceptable for publication.

Response: All minor line changes in Pdf accepted.

Reviewer: 2

Comments to the Author(s)

I thank the authors for accommodating my earlier comments into the present revision, resulting in a much improved MS. There are still some minor revisions needed; I have made numerous suggestions in the attached PDF. The phylogenetic analysis for some reason precedes the description, making for analysis for a taxon yet to be named! It should follow the syst paleo and description. Be sure that all abbreviations in the figures are defined in text (currently they are not).

Response: All minor line changes in Pdf accepted. Phylogenetic section moved to just before the Discussion. Abbreviations added to text.

My major scientific issue comes with the identification of the "supraoccipital". In the figure of the skull (which is slightly oblique of a full dorsal view) the midline suture appears to continue posterior of the parietals, and through the "supraoccipital", making it a paired structure, which would be odd indeed. I strongly encourage the authors to consider they instead have illustrated paired postparietals. If this interpretation is accepted then it will require some follow on alteration of the description and discussion and will probably impact upon their phylogenetic results.

Response: Currently we do not accept/agree with this interpretation, as mentioned in the text the skull is folded over on one side, thus only giving the appearance that the supraoccipital is at the midline. Where in reality it occupies the center of the range of occipital elements present, and it is also emarginated ventrally to form the dorsal surface of the foramen magnum! This is the best interpretation at the moment in our opinion.

Thank you for your current and previous revisions

-Arjan Mann